# Impact of Climate Change on the Distribution of Three Rare Salamanders (*Liua shihi*, *Pseudohynobius jinfo*, and *Tylototriton wenxianensis*) in Chongqing, China, and Their Conservation Implications

**DOI:** 10.3390/ani14050672

**Published:** 2024-02-21

**Authors:** Qi Ma, Lipeng Wan, Shengchao Shi, Zhijian Wang

**Affiliations:** 1State Key Laboratory Breeding Base of Eco-Environment and Bio-Resource of the Three Gorges Area, School of Life Sciences, Southwest University, Chongqing 400700, China; maqiswu@126.com (Q.M.); wanlipeng1216@126.com (L.W.); 2Chongqing Natural History Museum, Chongqing 400700, China; 3Hubei Engineering Research Center for Protection and Utilization of Special Biological Resources in the Hanjiang River Basin, School of Life Science, Jianghan University, Wuhan 430056, China; 4Chengdu Institute of Biology, Chinese Academy of Sciences, Chengdu 610041, China

**Keywords:** *Liua shihi*, *Pseudohynobius jinfo*, *Tylototriton wenxianensis*, MaxEnt, climate change, species conservation

## Abstract

**Simple Summary:**

Biodiversity conservation under global climate change is a global challenge. Many wildlife species are altering their distributions to adapt to the changes in their living environments caused by climate change. Amphibians are the most vulnerable group among vertebrates in the context of global climate change, and salamanders are one of the most vulnerable groups among amphibians. The primary threat to salamanders is habitat loss. There have been multiple studies on the potential distribution prediction of umbrella species (umbrella species are those species whose conservation is expected to provide protection for a wide range of co-occurring species due to their extensive habitat requirements), such as the Chinese giant salamander (*Andrias davidianus*). However, research on rare small salamanders is urgently needed for their conservation under climate change conditions. This study employed an optimized MaxEnt model to predict and analyze the potential distribution and trends of three rare salamanders from Chongqing, China, based on data collected from field surveys, museum collections, and the existing literature. This study assesses the impact of various environmental factors in the context of climate change on three salamanders species with different habitat preferences in Chongqing. It offers implications for the conservation of these salamanders.

**Abstract:**

The Wushan Salamander (*Liua shihi*), Jinfo Salamander (*Pseudohynobius jinfo*), and Wenxian Knobby Salamander (*Tylototriton wenxianensis*) are rare national Class II protected wild animals in China. We performed MaxEnt modeling to predict and analyze the potential distribution and trends of these species in Chongqing under current and future climate conditions. Species distribution data were primarily obtained from field surveys, supplemented by museum collections and the existing literature. These efforts yielded 636 records, including 43 for *P. jinfo*, 23 for *T. wenxianensis*, and 570 for *L. shihi*. Duplicate records within the same 100 m × 100 m grid cell were removed using ENMTools, resulting in 10, 12, and 58 valid distribution points for *P. jinfo*, *T. wenxianensis*, and *L. shihi*, respectively. The optimization of feature class parameters (FC) and the regularization multiplier (RM) were applied using R package “ENMeval 2.0” to establish the optimal model with MaxEnt. The refined models were applied to simulate the suitable distribution areas for the three species. The results indicate that the current suitable habitat area for *L. shihi* accounted for 9.72% of the whole region of the Chongqing municipality. It is projected that, by 2050, the proportion of suitable habitat will increase to 12.54% but will decrease to 11.98% by 2070 and further decline to 8.80% by 2090. The current suitable habitat area for *P. jinfo* accounted for 1.08% of the whole region of the Chongqing municipality, which is expected to decrease to 0.31%% by 2050, 0.20% by 2070, and 0.07% by 2090. The current suitable habitat area for *T. wenxianensis* accounted for 0.81% of the whole region of the Chongqing municipality, which is anticipated to decrease to 0.37% by 2050, 0.21% by 2070, and 0.06% by 2090. Human disturbance, climate variables, and habitat characteristics are the primary factors influencing the distribution of three salamander species in Chongqing. The proximity to roads significantly impacts *L. shihi*, while climate conditions mainly affect *P. jinfo*, and the distance to water sources is crucial for *T. wenxianensis*. The following suggestions were made based on key variables identified for each species: (1) For *L. shihi*, it is imperative to minimize human disturbances and preserve areas without roads and the existing vegetation within nature reserves to ensure their continued existence. (2) For *P. jinfo*, the conservation of high-altitude habitats is of utmost importance, along with the reduction in disturbances caused by roads to maintain the species’ ecological niche. (3) For *T. wenxianensis*, the protection of aquatic habitats is crucial. Additionally, efforts to mitigate the impacts of road construction and enhance public awareness are essential for the preservation of this species and the connectivity of its habitats.

## 1. Introduction

Amid intensifying climate change and human activities, global biodiversity is undergoing an unprecedented rapid decline. This not only leads to the disappearance of a multitude of species but also signifies the loss of precious evolutionary history and irreplaceable ecosystem services. Against this backdrop, the conservation of biodiversity and the prevention of species extinction have become urgent global imperatives [1]. Climate change has exerted a wide range of impacts on species, including alterations in their geographic distribution patterns, relative abundance, and phenology. The global rise in temperature, along with other environmental stressors, has affected over 80% of extant species [2]. Climate-induced changes in temperature and precipitation patterns have altered the natural habitats of wildlife, causing species distribution to shift in response to environmental changes [3].

Amphibians, bridging aquatic and terrestrial ecosystems, are extremely sensitive to minor environmental changes and are, thus, often used as indicator species for environmental variation. Due to their ectothermic nature, limited migratory capabilities, and strong dependence on water, amphibian species are at a higher risk of extinction compared to other taxa [4]. The conservation of amphibians and the prevention of their population decline are crucial for maintaining the health and stability of ecosystems [5,6,7,8]. In the context of climate change, given their limited abilities to migrate, more than half of the world’s amphibian species are predicted to lose over half of their current habitats by 2080 under a realistic dispersal scenario [9]. Globally, amphibians exhibit the highest vulnerability among vertebrates, with 40.7% of species at risk of extinction. Amphibian species in China are expected to lose an average of 20% of their existing habitats [10]. Among all extant groups of amphibians, salamanders are facing the most severe survival threats [11,12]. According to the data from the International Union for Conservation of Nature (IUCN), among the 759 assessed species of salamanders, a significant proportion, amounting to 56.52%, is categorized as vulnerable (VU) or higher on the endangerment scale. Within this subset, three species have been definitively identified as extinct [13]. Furthermore, salamanders constitute the majority (69%) of the 93 amphibian species listed as Class I or II nationally protected wildlife in China [14]. However, the geographic distribution of most amphibian species is still not known in detail. Research on the spatial distribution patterns of amphibian habitats, particularly those of salamanders, is crucial for the conservation of these species.

Climate change is a major driver of increasing extinction risk for amphibians. Amphibians are particularly vulnerable to climate change due to their aquatic and terrestrial life histories and low dispersal abilities, which make it difficult for them to adapt or move to more suitable habitats as conditions change [15]. Habitat loss represents the most prevalent threat to amphibian species, leading to the destruction or fragmentation of the environments upon which these species depend for their survival [16]. Pathogenic diseases such as the lethal chytrid fungi *Batrachochytrium dendrobatidis* (*Bd*) and *Batrachochytrium salamandrivorans* (*Bsal*) [17,18], exacerbated by climate change and habitat alteration, are responsible for rapid and widespread declines in amphibian populations, often driving species to the brink of extinction. The spread of pathogens such as the chytrid fungus has been linked to global amphibian declines, facilitated by environmental changes and human movement [19,20,21].

Given these challenges, it is important to inventory suitable habitats and potential distribution of amphibian species under climate change. The maximum entropy model has been extensively employed in the prediction of potential amphibian species’ distribution in various studies across the world [15,22,23,24]. Currently, research on potential distribution predictions for Chinese salamander species mainly focuses on umbrella species such as the Chinese giant salamander (*Andrias davidianus*) [25,26,27,28]. It is necessary to also investigate distribution predictions for rare small salamanders like the Wushan Salamander (*Liua shihi*), the Jinfo Salamander (*Pseudohynobius jinfo*), and the Wenxian Knobby Salamander (*Tylototriton wenxianensis*) (Figure 1).

These three small salamanders are rare species in China. They are designated as Class II nationally protected wildlife in accordance with the Law of the People’s Republic of China on the Protection of Wildlife, and they possess considerable importance for the conservation of biodiversity [14]. According to assessments by the International Union for Conservation of Nature (IUCN), *Pseudohynobius jinfo* is listed as endangered (EN), *Tylototriton wenxianensis* is listed as vulnerable (VU), and *Liua shihi* as being of least concern (LC) [29,30,31]. In the publication “*China’s Red List of Biodiversity: Vertebrates Volume IV Amphibians*”, *P. jinfo* is listed as being critically endangered (CR), *T. wenxianensis* as vulnerable (VU), and *L. shihi* as near threatened (NT) [11,12]. Furthermore, as of 2023, *T. wenxianensis* has been incorporated into Appendix II of the Convention on International Trade in Endangered Species of Wild Fauna and Flora (CITES) [32]. These three salamander species have different habitat preferences corresponding to their unique life histories, which makes them ideal subjects for the exploration of how different groups of salamanders react to human interference and habitat change under climate change.

The Wushan Salamander (*Liua shihi*), a member of the family Hynobiidae, has a limited and fragmented distribution spanning less than 20,000 km^2^, encompassing regions within the Henan province, the Sichuan province, the Chongqing municipality, and the Hubei province. Although this species is currently prevalent, there has been a noted decline in its population, attributable to the reduction in both their habitat area and quality as well as the impacts of overhunting [12]. Adult Wushan Salamanders are typically found in the aquatic environments of small mountain streams or within burrows adjacent to stream banks. Their larvae can be found beneath stones or within rock crevices along the banks of these streams, at elevations ranging from 900 to 2350 m [33]. The predominant vegetation in these habitats includes plant species such as the *Quercus fabri* and the *Quercus glandulifera*, among others [33,34].

The Jinfo Salamander (*Pseudohynobius jinfo*), belonging to the family Hynobiidae, has been documented exclusively within the regions of Chongqing and Guizhou [35,36]. The population size of this species is small and continues to decline [11]. During the breeding season, adult Jinfo Salamanders seek refuge in the grass adjacent to streams during daylight hours and are found in aquatic environments at night. In the nonbreeding season, they inhabit moist leaf litter beneath dense shrubbery and grass, distanced from streams, with their tadpoles residing in the puddles of minor streams. This species occupies an elevation range from 1700 to 2150 m [33]. The predominant vegetation within these habitats comprises thickets of the *Castanea* spp., *Quercus fabri*, and forests dominated by *Pinus massoniana*, among others [34].

The Wenxian Knobby Salamander (*Tylototriton wenxianensis*), a member of the Salamandridae family, is found across several regions in China, including the Gansu province, the Sichuan province, the Chongqing municipality, and the Guizhou province [11,33]. Despite its broad distribution, genetic analyses have revealed significant divergence among populations from different geographic locations, indicating that this species is actually a complex of multiple distinct species [37,38]. Consequently, the conservation of each population is critical to maintaining the species’ genetic diversity. The species’ habitat is highly fragmented, and its population has suffered a decline exceeding 30% over the past decade, primarily due to overharvesting for traditional medicine and the degradation of habitat quality and area resulting from pesticide and fertilizer use [11]. Adult Wenxian Knobby Salamanders lead a terrestrial existence within the dense underbrush of forests or shrublands and hibernate on land. During the breeding season, they inhabit marshy wetlands or still ponds, where their larvae develop. Their altitudinal range spans from 900 to 1400 m [33,39], and they are typically associated with vegetation communities that include species such as the *Miscanthus* spp., the *Juncus* spp., and the *Imperata* spp. [33,34,39].

The restricted geographic distribution of these three species, coupled with their heightened sensitivity to environmental conditions and the threats posed by environmental changes, infectious diseases, and unchecked development, has led to precipitous population declines, placing them at an imminent risk of extinction. It is urgent to draw a distribution map for the conservation of these species at present and under future climate change conditions.

From 2022 to 2023, building upon the foundation of the Second National Survey of Terrestrial Wildlife Resources from July 2014 to July 2018 in the Chongqing region of China, we initiated a series of specialized investigations focusing on three salamander species in Chongqing. We carefully selected representative sampling sites within the known distribution ranges of each species, ensuring a comprehensive dataset by including a spectrum of elevations, diverse aquatic habitats, and varied vegetation covers. Integrating data from museum collections and the extant literature on species distribution and taking into account a multitude of factors such as climatic conditions, topography, habitat quality, and human-induced disturbances, we utilized the refined MaxEnt model to forecast and scrutinize the potential distribution and trends of the species *Liua shihi*, *Pseudohynobius jinfo*, and *Tylototriton wenxianensis* within the Chongqing region. By evaluating the influence of various environmental determinants and projecting the distribution of these three salamanders—each with distinct habitat proclivities—under scenarios of climate change, our research provides critical insights for the conservation of these endemic salamanders.

## 2. Materials and Methods

### 2.1. Study Area

Chongqing is a significant municipality in the southwest of China, alongside Beijing, Shanghai, and Tianjin, holding the same administrative status as a province. It is essential to clarify that, within the broader municipal boundaries of Chongqing, there exists a core urban district also identified as Chongqing. For the purposes of this paper, the term “Chongqing” is employed to denote the extensive municipal region, rather than the narrower confines of the central urban district (Figure 2). Geographically, it extends from 105°17′ E to 110°11′ E in longitude and from 28°10′ N to 32°13′ N in latitude, encompassing a vast expanse of approximately 82,400 km^2^. Chongqing’s dimensions stretch to 470 km from east to west and span 450 km from north to south [40]. Situated in the subtropical monsoon climate zone, Chongqing experiences a mild climate throughout the year, with distinct seasons, abundant precipitation, high humidity, overlapping periods of heat and rain, limited sunshine, low wind speeds, and frequent fog. Most regions have an average annual temperature ranging from 13 to 19 °C, which is higher than other areas at the same latitude in China [41]. The area is rich in vertebrate species, including many which are under national protection.

Given its heterogeneous landscape, the Chongqing municipality is a good region to study the influences of habitat change for salamanders under different climate change scenarios. Chongqing’s varied topography is primarily dominated by mountains and hills, with higher elevations in the northeast and southeast, while the central and western parts are lower, featuring steep slopes. The region is well-drained, with the mainstream of the Yangtze River and its tributaries originating from the mountainous areas to the north and south of Chongqing. The northeastern region of Chongqing is characterized by the Daba Mountains and the Three Gorges valley terrain, with elevations ranging from 500 to 2800 m. The predominant vegetation types in this area are subtropical evergreen and deciduous broadleaf shrubs, cultivated plants, and subtropical and tropical grasslands. In the central, western, and northern parts, the landscape is primarily composed of mountainous and hilly terrain with elevations below 1200 m, where cultivated plants and subtropical coniferous forests are the main vegetation types. The northwestern part is dominated by plain terrain with elevations below 1000 m, and the primary vegetation type is cultivated plants. The southeastern part is characterized by mid-mountain canyon terrain and mountainous terrain with elevations above 400 m, where the main vegetation types are subtropical and tropical evergreen broadleaf, deciduous shrubs, and cultivated plants [34].

The Chongqing municipality exhibits a high human population density, yet the population distribution is uneven; this provides an opportunity to test human influence on salamanders. According to data from 2022, the population density reached 414 individuals per square kilometer [42]. Centering around the Yuzhong district, the human population is distributed radially outward, with density gradually decreasing. Among these areas, the central Yuzhong district boasts the highest human population density, reaching 28,239 individuals per square kilometer in 2017; in contrast, the northeastern Chengkou County has the lowest density, at only 57 individuals per square kilometer [43]. Such an uneven human population distribution may exert direct or indirect impacts on the survival and habitats of salamanders. Since becoming a municipality in 1997, it has faced increasingly severe biodiversity loss amidst rapid economic growth and urbanization. Understanding the current distribution of rare salamander species in Chongqing, the primary environmental factors affecting their survival, and the trends in habitat suitability now and in the future are of great significance for local conservation strategies and future planning. This knowledge can provide vital support for the protection of biodiversity and sustainable development.

### 2.2. Occurrence Data Collection

#### 2.2.1. Field Surveys

From July 2014 to July 2018, we conducted field surveys from the Second National Survey of Terrestrial Wildlife Resources in the Chongqing municipality, which collected basic distribution data of terrestrial vertebrates including amphibians. A total of 1760 transects distributed in 88 square grids (10 km × 10 km) uniformly covering Chongqing were surveyed, with 20 transects in each grid (Figure 3). The field surveys were conducted during summer months (June and July) and winter months (January and February). Each transect was 3 km long, and each was separated from the others by a distance of at least 1 km.

In February–March 2022, July–August 2022, and April–August 2023, we conducted surveys targeting these three salamander species in Chongqing. Drawing upon data amassed from field surveys conducted between 2014 and 2018 as well as natural history data extracted from the literature, we identified and subsequently excluded areas unsuitable for the habitation of the three species under consideration, primarily those plains situated at altitudes below 500 m. Consequently, we selected the northeastern Daba Mountains and the Wushan County Mountains, the eastern Qiyao Mountains and the Fangdou Mountains, and the southern Dalou Mountains’ region within the Chongqing municipality as the principal areas for our survey. The surveyed areas included nature reserves, forest parks, and wetland parks. The feasibility of survey routes was also considered, and unstructured interviews were conducted with the forestry bureau and reserve staff, forest rangers, and villagers to collect extensive and reliable distribution data. A total of 290 transects were surveyed (Figure 3), covering habitats of the three salamander species, including streams and permanent or seasonal ponds.

The length of each survey transect varied with the terrain, typically ranging from 1 to 3 km, with a width of 15 m on one side and spaced more than 1 km apart. The surveys were conducted on foot along the transects at an average speed of about 1.6 km/h, primarily in the afternoon until late at night, from 3:00 p.m. (15:00) until 1:00 a.m. the following day. The survey teams typically consisted of two people or, occasionally, three to five. During the surveys, cover objects within the transect were overturned to search for salamanders hiding underneath, and dip nets were used to search for larvae in water bodies or streams that were difficult to observe. The presence or absence of the three species, the main vegetation type, longitude, latitude, altitude, and temperature were recorded for each transect. Photographs were also taken of the salamanders for identification. As salamanders are nationally protected animals, we did not collect specimens. For some larvae which were difficult to identify, non-invasive DNA sampling methods were employed [44], and DNA samples were sequenced for molecular identification.

#### 2.2.2. Museum and Literature-Based Distribution Data Collection

To obtain more comprehensive information on species distribution and natural history, we examined historical specimens of the three salamander species from Chongqing preserved in the Herpetological Museum of the Chengdu Institute of Biology, Chinese Academy of Sciences (CIB CAS), the Zoological Museum of Kunming Institute of Zoology, Chinese Academy of Sciences (KIZ CAS), the Zoological Museum of Southwest University (SWU), and the Chongqing Museum of Natural History (CMNH) in May–June 2022. Finally, we obtained three distribution records for *Pseudohynobius jinfo* and twenty for *Liua shihi* from the museum specimens. Additional occurrence data for *Tylototriton wenxianensis* were derived from the published literature [39]. In total, 636 geographical distribution records were acquired, including 43 for *P. jinfo*, 23 for *T. wenxianensis*, and 570 for *L. shihi*.

#### 2.2.3. Occurrence Data Processing

The geographic coordinate system of the collected occurrence data was transformed from WGS_1984 to the WGS_1984_UTM_Zone_49N projected coordinate system, which is suitable for the Chongqing area. To prevent overfitting, duplicate distribution sites of the same species located within the same grid cell (100 m × 100 m) were removed using ENMTools.pl (https://github.com/danlwarren/ENMTools, accessed on 3 October 2023) [45]. After processing the data as described above, we identified 10 valid occurrence sites for *Pseudohynobius jinfo*, 12 for *Tylototriton wenxianensis*, and 58 for *Liua shihi* (Figure 4). The occurrence site numbers were larger than nine, which enabled the following analysis with an acceptable predictive accuracy [46]. The occurrence site number seems relatively small, but this does not imply an inadequate characterization of the actual ecological niche of the species distribution. Rather, it is due to the inherently narrow distribution ranges of these endemic amphibian species. In such cases, even increasing the number of distribution points would only augment the density of points within the existing regions and would not significantly enhance the model’s predictive performance [15,47].

### 2.3. Environmental Variables

Based on the biological characteristics of amphibian species, we selected twenty-six environmental variables (Table 1), which included nineteen climatic factor variables, three topographic factor variables (elevation, slope, aspect), two habitat factor variables (distance to water sources, normalized difference vegetation index), and two disturbance factor variables (distance to roads, distance to human settlements).

Nineteen bioclimatic layers were sourced from the WorldClim dataset (https://www.worldclim.org , accessed on 1 October 2023), selecting climate data from the years 1970–2000 to represent the current climate conditions [48,49]. Future climate projections utilized the WorldClim version 2.1 (https://www.worldclim.org, accessed on 1 October 2023), with a resolution of 30 arc-seconds. To predict changes in suitable habitat for the years 2041–2060 (hereafter referred to as 2050), 2061–2080 (2070), and 2081–2100 (2090), we selected the mean values of four Shared socioeconomic pathways (SSPs) under the BCC-CSM2-MR climate model [50]. The SSPs include SSP126 (low greenhouse gas emissions), SSP245 and SSP370 (medium greenhouse gas emissions), and SSP585 (high greenhouse gas emissions). In addition to bioclimatic variables, we also considered topographic, habitat, and anthropogenic disturbance factors, including elevation, slope, aspect, distance to the nearest road, distance to the nearest residential area, distance to the nearest surface water, and the normalized difference vegetation index (NDVI).

Digital Elevation Model (DEM) data were sourced from the Geospatial Data Cloud platform of the Computer Network Information Center, Chinese Academy of Sciences (https://www.gscloud.cn, accessed on 1 October 2023); slope and aspect were calculated using ArcGIS 10.4.1 (ESRI Inc.). Data of residential sites and roads were acquired from the National Geomatics Center of China (https://www.webmap.cn, accessed on 1 October 2023). A residential site primarily includes residential areas, ordinary houses, grazing sites, etc., while road data encompass railways, expressway, provincial roads, county roads, village roads, streets, and rural roads. Water source factor data were also obtained from the National Geomatics Center of China. The Euclidean Distance function in ArcGIS was utilized for distance analysis. NDVI data were derived from the National Science and Technology Infrastructure (http://www.nesdc.org.cn, accessed on 1 October 2023), selecting the average of the 30 m annual maximum NDVI dataset for China from 2000 to 2020 [51]. Finally, the pixel size for all environmental variables (X, Y) was uniformly set to (100, 100), and the coordinate system used was the UTM_WGS_1984_Zone_49N projection coordinate system suitable for the Chongqing area.

To mitigate the negative impact of a high spatial collinearity among environmental factors on the model predictions and enhance the accuracy of the forecasted results, environmental factor screening was conducted for each species. Initially, a pre-experiment was performed for each species using the MaxEnt version 3.4.4 software (New York, NY, USA, https://biodiversityinformatics.amnh.org/open_source/maxent/, accessed on 1 October 2023). Subsequently, a correlation analysis of all environmental factors was carried out using the correlation function in the ENMTools.pl [45], and correlation coefficients were calculated (Figure 5). If two variables exhibited a high correlation (|r| ≥ 0.8), the environmental factor with the lower percent contribution in the analysis of variable contributions from the initial MaxEnt model experiment was removed. Following these steps, eight environmental factors were retained for the final modeling (Appendix A).

### 2.4. Model Parameter Optimization

The MaxEnt model is based on the principle of maximum entropy, which seeks the probability distribution with the highest entropy from the given conditions [52]. It assumes that the most realistic species distribution should have the highest entropy probability distribution [53]. To obtain the species distribution required for the model, specific locations (latitude and longitude coordinates) of the species are obtained, and the relevant environmental factors of the species are used as constraints to limit its distribution range. By establishing connections to analyze the distribution range with maximum entropy under these constraints, the suitable habitat distribution area for the species can be inferred [53,54]. Even with a small sample size, the MaxEnt model remains robust because its feature functions can be adjusted according to the sample size [55,56,57]. For future predictions, MaxEnt can incorporate scenarios of climate change using projections of future climate conditions as the input variables [53]. These projections are typically derived from climate models and represent different potential future climates under various greenhouse gas emission scenarios [58,59]. By applying these future climate scenarios to the MaxEnt model, researchers can predict how the distribution of species may shift in response to changing climatic conditions [60,61,62,63,64].

MaxEnt has demonstrated a high predictive performance in species distribution modeling and is widely applied [28,65,66,67,68]. However, to avoid the generation of misleading results, meticulous optimization is required. By adjusting the combination of feature class parameters (FC) and the regularization multiplier (RM), the model can be customized according to the specific circumstances of the species, thereby enhancing the model’s accuracy and flexibility [66,69,70,71].

The R package ”ENMeval 2.0” [72] was used to optimize the regularization multiplier and feature class parameters within the R software version 4.3.2 (Vienna, Austria, https://www.r-project.org/, accessed on 1 October 2023). These two parameters are crucial for constructing species distribution models using the MaxEnt software version 3.4.4. Specifically, our study set the regularization multiplier range from 0.5 to 4, increasing by increments of 0.5, and tested six feature class parameters—namely, “H” for hinge, “L” for linear, “LQ” for linear quadratic, “LQH” for linear quadratic hinge, “LQHP” for linear quadratic hinge product, and “LQHPT” for linear quadratic hinge product threshold [73]—resulting in a total of 48 parameter combinations for testing [72]. Model selection was based on the delta akaike information criterion (Delta AICc) corrected for a small sample size, the area under the curve for the training data (AUC.train), the continuous boyce index for the training data (CBI.train), the average difference in the AUC between the training and testing datasets (AUC.diff.avg), and the 10% omission rate (or.10p.avg) [74,75]. Among these metrics, the model with the smallest delta AICc value (delta AICc = 0) was identified as the optimal model [68].

### 2.5. Species Distribution Models

Our species distribution models were constructed using MaxEnt, and projections were made for three future time periods. Each species’ model was parameterized with distinct regularization multipliers and feature class parameters. To ascertain the precision of the predictive models, a quarter of the sample entries were designated as a “random test percentage”. This allocation entailed the algorithm randomly segregating 25% of the data entries to serve as a test subset. Consequently, the calibration (training) phase utilized 75% of the dataset, while the evaluation (testing) phase employed the remaining 25%. To enhance the reliability of the results, ten replicates were conducted within the study. The relative contribution of each environmental variable was assessed using the jackknife test.

### 2.6. Changes in Suitable Area under Different Climatic Conditions

In the final model, the suitability index ranged from 0 to 1, representing the potential degree of suitability for the species. To distinguish between suitable and unsuitable habitats, we utilized a specific threshold derived from the model’s output. This threshold is based on the Maximum Training Sensitivity Plus Specificity (MTSPS) criterion, which is a value selected to maximize the sum of the model’s sensitivity and specificity. The MTSPS threshold is used to convert the continuous probability of a species’ presence into a binary classification of suitable and unsuitable habitats [61,64,76,77,78]. Combining the distribution data and the fit of the species’ suitable areas, the criteria for classifying habitat suitability based on the probability of presence were as follows: *Liua shihi* MTSPS = 0.0557; *Pseudohynobius jinfo* MTSPS = 0.1545; and *Tylototriton wenxianensis* MTSPS = 0.5285. Therefore, the suitable area for *L. shihi* was where *p* > 0.0557, and the unsuitable area was where *p* ≤ 0.0557; for *P. jinfo*, the suitable area was where *p* > 0.1545, and the unsuitable area was where *p* ≤ 0.1545; for *T. wenxianensis*, the suitable area was where *p* > 0.5285, and the unsuitable area was where *p* ≤ 0.5285.

The simulation results utilized the SDMtool to calculate the changes in the distribution center points and areas under different temporal scenarios [79]. The SDMtoolbox was loaded in ArcGIS and the “Distribution Changes Between Binary SDMs” tool from the “Universal Tools” subdirectory was used to calculate the predicted results for *Liua shihi*, *Pseudohynobius jinfo*, and *Tylototriton wenxianensis* at different times, converting the results into binary grid files. Subsequently, the “Centroid Changes (Lines)” tool was employed to calculate the geometric centroid shifts of the predicted distributions over different periods and detect the overall trend of change in the suitable areas for *L. shihi*, *P. jinfo*, and *T. wenxianensis*.

## 3. Results

### 3.1. Optimal Model and Model Accuracy Evaluation

Under the default parameters of MaxEnt (RM = 1 and FC = LQHP), the delta AICc for *Liua shihi* was 74.726, with an AUC of 0.988, a CBI of 0.973, an AUC.diff of 0.013, and an OR of 0.155. When the parameters were set to RM = 0.5 and FC = LQ, the delta AICc was 0, the AUC was 0.984, the CBI was 0.978, the AUC.diff was 0.018, and the OR was 0.103 (Figure 6). With the default MaxEnt parameters (RM = 1 and FC = LQHP), the delta AICc for *Pseudohynobius jinfo* was not available, the AUC was 0.998, the CBI was 0.777, the AUC.diff was 0.004, and the OR was 0.300. When the parameters were RM = 3.5 and FC = LQH, the delta AICc was 0, the AUC was 0.998, the CBI was 0.565, the AUC.diff was 0.004, and the OR was 0.300. Under the default MaxEnt parameters (RM = 1 and FC = LQHP), the delta AICc for *Tylototriton wenxianensis* was not available, the AUC was 0.998, the CBI was 0.721, the AUC.diff was 0.014, and the OR was 0.167. When the parameters were RM = 3.5 and FC = LQH, the delta AICc was 0, the AUC was 0.997, the CBI was 0.806, the AUC.diff was 0.021, and the OR was 0.167.

Through the optimization of the parameters, the complexity of the model was significantly reduced, making it more suitable for migration modeling during different periods. Consequently, we selected RM = 0.5 and FC = LQ as the modeling parameters for the ecological niche simulation of *Liua shihi*, RM = 3.5 and FC = LQH for *Pseudohynobius jinfo*, and RM = 3.5 and FC = LQH for *Tylototriton wenxianensis* (see Appendix A). The chosen parameter combinations were used to simulate and predict the current and future distributions of the species. The ROC curve validation results for *L. shihi*, *P. jinfo*, and *T. wenxianensis* indicated that the average AUC values from ten repetitions were 0.981, 0.998, and 0.996, respectively, all of which were higher than the random AUC value (0.5) and close to 1.0, with CBI values greater than 0.5, indicating an excellent model with a good fit and a high predictive accuracy.

### 3.2. Key Environmental Factors of the Predicted Distribution of Liua shihi, Pseudohynobius jinfo, and Tylototriton wenxianensis

The results from the analysis of variable contributions in the MaxEnt model (see Appendix A) indicated that the key variable affecting the habitat quality of *Liua shihi* was the distance to the nearest road (74.7%), followed by the mean temperature of the coldest quarter (13.6%), isothermality (5.5%), the precipitation of the wettest month (2.2%), and elevation (2.1%). The cumulative contribution of these five environmental variables reached 98.1%, with human activities being the most significant influencing factor, accounting for 74.7%, followed by the climatic variables (21.3%) and the topographic variables (2.1%).

For *Pseudohynobius jinfo*, the key variable impacting habitat quality was the mean temperature of the wettest quarter (61.7%), followed by the distance to the nearest road (21.2%), the mean diurnal range (15.1%), the max temperature of the warmest month (1.1%), and the distance to the nearest surface water (0.7%). The cumulative contribution of these five environmental variables was 99.8%, with the climatic variables playing the most crucial role, accounting for 77.9%, followed by human activities (21.2%) and the habitat variables (0.7%).

For *Tylototriton wenxianensis*, the key variable affecting the quality of its habitat was the distance to the nearest surface water (58.2%), followed by the distance to the nearest road (30.5%), elevation (4.5%), the distance to the nearest residential area (2.4%), and the mean temperature of the coldest quarter (2.1%). The cumulative contribution of these five environmental variables reached 97.7%, with the habitat variables playing the most significant role, accounting for 58.2%. These were followed by the human activities’ variables (32.9%) and the topographic variables (4.5%).

### 3.3. Potential Suitable Areas for Liua shihi, Pseudohynobius jinfo, and Tylototriton wenxianensis under Current and Future Climate Conditions

*Liua shihi* is primarily distributed in the northeastern part of Chongqing, including Chengkou County, Wuxi County, Wushan County, Fengjie County, and the Kaizhou District. Currently,, the suitable habitat for *L. shihi* was distributed in 9.72% of the Chongqing municipality, while 90.28% of the municipality was unsuitable for this species (Figure 7a). The projections indicate that, by 2050, the suitable habitats will increase to 12.54%, and the unsuitable areas will decrease to 87.46% (Figure 7b). By 2070, the suitable habitats will decrease to 11.98%, and the unsuitable areas will slightly rise to 88.02% (Figure 7c). By 2090, the suitable habitats will diminish to 8.80%, and the unsuitable areas will further increase to 91.20% (Figure 7d, Appendix A).

*Pseudohynobius jinfo* is mainly found in the southern parts of Chongqing, specifically in the Nanchuan District and the Wulong District. At present, a suitable habitat for *P. jinfo* was distributed in 1.08% of the Chongqing municipality, while 98.92% of the municipality was unsuitable for this species (Figure 8a). According to predictions, by 2050, the suitable habitats will decrease to 0.31%, and the unsuitable areas will rise to 99.69% (Figure 8b). By 2070, the suitable habitats will reduce to 0.20%, and the unsuitable areas will continue to increase to 99.80% (Figure 8c). By 2090, the suitable habitats will decrease to 0.07%, and the unsuitable areas will significantly increase to 99.93% (Figure 8d, Appendix A).

*Tylototriton wenxianensis* is primarily distributed in the northeastern regions of Chongqing, specifically in the Yunyang County and the Fengjie County. Currently, a suitable habitat for *T. wenxianensis* was distributed in 0.81% of the Chongqing municipality, while 99.19% of the municipality was unsuitable for this species (Figure 9a). According to predictions, by the year 2050, the proportion of suitable habitats will decrease to 0.37%, with the unsuitable habitats being expected to increase to 99.63% (Figure 9b). By 2070, the suitable habitats will be reduced to 0.21%, while the unsuitable areas are projected to further increase to 99.79% (Figure 9c). By 2090, the suitable habitats will decrease to a mere 0.06%, while the unsuitable areas are anticipated to significantly increase to 99.94% (Figure 9d, Appendix A).

### 3.4. Spatial Transfer Characteristics of Suitable Areas for Liua shihi, Pseudohynobius jinfo, and Tylototriton wenxianensis

From the present to the year 2050 (Figure 10a), the expansion areas of suitable habitats for *Liua shihi* in Chongqing are primarily located in the northeastern parts, such as the southwestern part of the Chengkou County, the central and southern parts of the Wuxi County, the western part of the Wushan County, the northeastern and southeastern parts of the Fengjie County, the northeastern part of the Yunyang County, and the northeastern part of the Kaizhou District. Conversely, the contraction areas of suitable habitats are mainly concentrated in the central and eastern parts of the Wuxi County, the central and northern regions of the Wushan County, and a small northeastern section of the Fengjie County. The suitable habitat for this species is projected to expand by 2391.31 km^2^ and contract by 33.49 km^2^ in different areas. From 2050 to 2070 (Figure 10b), the expansion areas of suitable habitats for *L. shihi* in Chongqing will be mainly located in the central and northern parts of the Wushan County, whereas the contraction areas will be principally situated in the southern part of the Wuxi County, the central-western region of the Wushan County, and the northeastern sections of the Fengjie County, the Yunyang County, and the Kaizhou District. The suitable habitat is expected to expand by 10.54 km^2^ and contract by 478.25 km^2^. From 2070 to 2090 (Figure 10c), the expansion areas of suitable habitats for *L. shihi* in Chongqing will mainly be located in the central part of the Wushan County, while the contraction areas will mainly be in the central-western part of the Chengkou County, most parts of the Wuxi County, most parts of the Wushan County, the southeastern and northeastern parts of the Fengjie County, the northeastern part of the Kaizhou District, and the Yunyang County. The suitable habitat is forecasted to expand by 0.48 km^2^ and contract by 2658.82 km^2^ (Appendix A).

Due to the irregular shape of the suitable distribution area, the centroid is used to define the central point of the distribution area to characterize the shift in the location of the suitable habitat for this species under different climate change scenarios [80]. The centroid shift map (Figure 10d) indicates that the centroid of the suitable distribution area for *Liua shihi* will migrate southeastward from the present (109.2090° E, 31.6122° N, altitude 1810 m) to the year 2050 (109.2213° E, 31.5519° N, altitude 1587 m), with a migration distance of 6.81 km; from 2050 to 2070 (109.2099° E, 31.5680° N, altitude 1103 m), it will migrate northwestward, with a migration distance of 2.09 km; and, from 2070 to 2090 (109.1191° E, 31.6655° N, altitude 1888 m), it will continue to migrate northwestward with a migration distance of 13.84 km. Overall, the centroid migration exhibits a trend of initially moving southeastward (2050) and then northwestward (2070, 2090). The Wuxi County is the current and future distribution center for *L. shihi* in Chongqing.

From the present to the year 2050 (Figure 11a), the expansion areas of suitable habitats for *Pseudohynobius jinfo* in Chongqing are mainly located in a small southern region, including the central and southwestern parts of the Nanchuan District and the southwestern area of the Wulong District. Conversely, the contraction areas of suitable habitats are primarily in the central and southern regions of Chongqing, encompassing the southwestern parts of the Shizhu County, the Fengdu County, and the Pengshui County, the southeastern part of the Fuling District, the northern and southwestern parts of the Wulong District, the northeastern, central, and southwestern parts of the Nanchuan District, and the eastern part of the Qijiang District. It is projected that the suitable habitat area will expand by 7.16 km^2^ and contract by 639.14 km^2^. From 2050 to 2070 (Figure 11b), there will be no expansion areas for *Pseudohynobius jinfo* in Chongqing. The contraction areas of suitable habitats will mainly be located in the central and southern parts of Chongqing, including the southern part of the Fengdu County, the southeastern part of the Fuling District, the northwestern and southwestern parts of the Wulong District, the northeastern, central, and southwestern parts of the Nanchuan District, and the eastern part of the Qijiang District. The suitable habitat of this species is projected to contract by 89.44 km^2^. From 2070 to 2090 (Figure 11c), *P. jinfo* is not expected to expand into any suitable areas in Chongqing; instead, the suitable habitats are projected to contract mainly in the central and southern districts and counties of Chongqing, encompassing the southern part of the Fengdu County, the northwestern and southwestern parts of the Wulong District, the eastern, central, and southwestern parts of the Nanchuan District, and the eastern part of the Qijiang District. The suitable habitat area is expected to shrink by 104.81 km^2^ (Appendix A).

The suitable distribution centroid (Figure 11d) for *Pseudohynobius jinfo* is projected to shift southwest from the present (107.4536° E, 29.2548° N, altitude 753 m) to the year 2050 (107.3426° E, 29.1130° N, altitude 1254 m), with a migration distance of 19.10 km. From 2050 to 2070 (107.2951 ° E, 29.0661° N, altitude 1096 m), the centroid is expected to move southwestward, covering a distance of 6.96 km. Subsequently, from 2070 to 2090 (107.2156° E, 29.007° N, altitude 2078 m), the centroid is predicted to shift southwest, with a migration distance of 10.15 km. Overall, the centroid migration displays a trend of shifting towards the southwest direction. The Wulong District (current) and the Nanchuan District (2050, 2070, 2090) are identified as the distribution centers for *P. jinfo* in Chongqing.

From the present to the year 2050 (Figure 12a), the expansion areas of suitable habitats for *Tylototriton wenxianensis* in Chongqing are mainly located in the central and eastern parts of the Fengjie County and a small portion of the western region of the Wushan County. The contraction of suitable areas will mainly occur in the eastern part of the Yunyang County, the central region of the Fengjie County, and the western area of the Wushan County. It is projected that the suitable habitat area of this species will expand by 0.34 km^2^ and contract by 367.49 km^2^. From 2050 to 2070 (Figure 12b), there will be no expansion of suitable habitats for *T. wenxianensis* in Chongqing, with the contraction of suitable areas primarily in the eastern part of the Yunyang County, the central region of the Fengjie County, and the western and southwestern areas of the Wushan County, leading to a further reduction of 139.42 km^2^ in suitable habitats. From 2070 to 2090 (Figure 12c), there will be no expansion of suitable habitats for *T. wenxianensis* in Chongqing, and the contraction of suitable areas will mainly be located in the eastern part of the Yunyang County, the central region of the Fengjie County, and the western and southwestern sectors of the Wushan County, with a decrease of 124.85 km^2^ in suitable habitats (Appendix A).

The centroid of the suitable distribution area (Figure 12d) for *Tylototriton wenxianensis* is projected to shift southeastward from its current position (109.3525° E, 30.9309° N, altitude 546 m) to a new one in the year 2050 (109.3665° E, 30.9272° N, altitude 738 m), covering a distance of 1.40 km. From 2050 to 2070 (109.3741° E, 30.9143° N, altitude 972 m), the centroid is expected to move southeastward, traveling a distance of 1.61 km. Subsequently, from 2070 to 2090 (109.3736° E, 30.9091° N, altitude 1160 m), the centroid will continue to shift southwestward, with a migration distance of 0.58 km. Overall, the centroid migration for this species exhibits a trend of initially moving southeastward by the years 2050 and 2070, followed by a southwestward direction by the year 2090. The Fengjie County serves as the current and future distribution center for *T. wenxianensis* in Chongqing. These trends in centroid migration reflect the varying adaptive capacities of different species to climate change.

### 3.5. Overlapping Distribution Regions of Liua shihi, Pseudohynobius jinfo, and Tylototriton wenxianensis during Various Periods

Due to differences in habitat preferences, the predicted current and future suitable areas for *Liua shihi*, *Pseudohynobius jinfo*, and *Tylototriton wenxianensis* exhibit species specificity, with limited overlapping potential distribution areas (Figure 13), and the predictions are consistent with expectations.

Currently (Figure 13a), the suitable areas for *Liua shihi* are located in the regions of the Chengkou County, the Wushan County, and the Wuxi County, as well as the northeastern part of the Kaizhou District, the northern part of the Yunyang County, and the northern region of the Fengjie County in Chongqing. The suitable areas for *Pseudohynobius jinfo* encompass areas in the Shizhu County, a small southwestern portion of the Pengshui County, the southern part of the Fengdu County, the southeastern region of the Fuling District, the northern and southern parts of the Wulong District, the northeastern, central, and southwestern parts of the Nanchuan District, and the eastern area of the Qijiang District. The suitable areas for *Tylototriton wenxianensis* are found in parts of the eastern Yunyang County, the central Fengjie County, and the western Wushan County. The common potential distribution areas for *L. shihi* and *T. wenxianensis* are located in the western part of the Wushan County. *P. jinfo* and *L. shihi* as well as *P. jinfo* and *T. wenxianensis* do not share common potential distribution areas.

By 2050 (Figure 13b), the potential distribution areas for *Liua shihi* include the majority of the Chengkou County, the Wushan County, and the Wuxi County, as well as the northeastern part of the Kaizhou District, the northern part of the Yunyang County, and the northern and eastern parts of the Fengjie County in Chongqing. The potential distribution areas for *Pseudohynobius jinfo* may cover a small part of the southern Fengdu County, the southeastern part of the Fuling District, the northwestern and southwestern parts of the Wulong District, the northeastern, central, and southwestern parts of the Nanchuan District, and the eastern part of the Qijiang District. The eastern part of the Yunyang County, the central part of the Fengjie County, and the western part of the Wushan County have been identified as potential distribution areas for *Tylototriton wenxianensis*. Moreover, the western part of the Wushan County has been recognized as the shared potential distribution area for both *L. shihi* and *T. wenxianensis*. *P. jinfo* and *L. shihi* as well as *P. jinfo* and *T. wenxianensis* do not share common potential distribution areas.

By 2070 (Figure 13c), the potential distribution areas for *Liua shihi* are projected to include the majority of the Chengkou County, the Wushan County, and the Wuxi County, as well as the northeastern part of the Kaizhou District, the northern part of the Yunyang County, and the northern and eastern parts of the Fengjie County in Chongqing. The potential distribution areas for *Pseudohynobius jinfo* are anticipated to encompass the southern part of the Fengdu County, the northwestern and southwestern parts of the Wulong District, the eastern, central, and southwestern parts of the Nanchuan District, and the eastern part of the Qijiang District. The potential distribution areas for *Tylototriton wenxianensis* are expected to be located in the eastern part of the Yunyang County, a small part of the western Wushan County, and the central part of the Fengjie County. There are no shared potential distribution areas for *L. shihi*, *P. jinfo,* and *T. wenxianensis*.

By 2090 (Figure 13d), the potential distribution areas for *Liua shihi* are projected to encompass significant portions of the Chengkou County and the Wushan County, as well as the northeastern, southeastern, and central-western parts of the Wuxi County, the northeastern part of the Kaizhou District, the northern part of the Yunyang County, and small sections of the northern and eastern parts of the Fengjie County. The potential distribution areas for *Pseudohynobius jinfo* are expected to include small sections of the southwestern part of the Wulong District and the eastern, central, and southwestern parts of the Nanchuan District. The eastern part of the Yunyang County, small sections of the western part of the Wushan County, and the central part of the Fengjie County have been identified as a potential distribution area for *Tylototriton wenxianensis*. There are no common potential distribution areas for *L. shihi*, *P. jinfo*, and *T. wenxianensis* (Figure 13d).

## 4. Discussion

### 4.1. Impact of Anthropogenic Disturbances, Bioclimatic Variables, and Water Sources on the Predicted Distribution Areas of Liua shihi, Pseudohynobius jinfo, and Tylototriton wenxianensis

Anthropogenic activities have a significant impact on amphibian populations, contributing to habitat alteration, fragmentation, and loss, which are major threats to these species [15]. Roads and traffic can cause habitat and population fragmentation, leading to higher rates of roadkill among amphibians, particularly as they migrate between wetland and upland habitats [81]. Moreover, the development of roads increases the likelihood of occurrences of small populations. The segmentation of habitats into smaller, discrete enclaves due to road construction can disunite portions of wildlife populations, leading to a diminution in genetic diversity and a contraction in the size of established populations, thereby precipitating an elevated risk of extinction [82,83,84]. Results from our environmental factor analyses indicate that, for *Liua shihi*, the distance to the nearest road (DRO) is the most significant variable, contributing 74.7% to the distribution prediction, while the distance to the nearest residential area (DRE) only contributes 1%. For *Pseudohynobius jinfo*, DRO still plays a substantial role, with a 21.2% contribution. Lastly, for *Tylototriton wenxianensis*, DRO is again a major factor, with a 30.5% contribution, and DRE has a smaller but notable contribution of 2.4% (Appendix A). The construction of roads may directly damage their habitats, including the pollution of aquatic environments, the fragmentation of habitats, and interference with their behavioral patterns. Additionally, the noise generated by roads could affect their reproductive behaviors, while light pollution may impact their foraging strategies and predator avoidance tactics. Compared to DRO, the contribution of DRE is lower, which may be related to the habitat environments of the species. *L. shihi* has a broad habitat distribution in Chongqing, including areas with fewer residential zones, such as protected areas, and also regions closer to residential areas. *P. jinfo* is currently only found within protected areas, with virtually no surrounding residential settlements. *T. wenxianensis* breeds in habitats such as wells, rice paddies, ponds, and wetlands, which are permanent or temporary still water bodies. Its distribution in Chongqing is not confined to protected areas and is closer to residential areas, thus experiencing greater human disturbance, which is reflected in the relatively higher contribution of DRE compared to the other two species.

Climate change has altered regional temperature and precipitation patterns, resulting in changes to animal habitats and even the loss of habitats. By analyzing the impact of bioclimatic variables on the distribution of the three species—*Liua shihi*, *Pseudohynobius jinfo,* and *Tylototriton wenxianensis*—we can observe that these variables play varying roles for each species. For *L. shihi*, the Bio11 variable, which represents the mean temperature of the coldest quarter, has a significant contribution of 13.6% to the species’ distribution prediction. This suggests that the coldest quarter’s temperature is an important factor for the habitat suitability of *L. shihi*. Isothermality and the precipitation of the wettest month (Bio13) also contribute to the model, albeit to a lesser extent, with 5.5% and 2.2%, respectively. In the case of *P. jinfo*, the mean temperature of the wettest quarter is the most influential bio variable, with a substantial contribution of 61.7%. This indicates that the temperature during the wettest quarter is critical for the distribution of this species. The mean diurnal range also has a notable contribution of 15.1%, suggesting that the temperature range within a day is also a significant factor for this species’ distribution. For *T. wenxianensis*, the Bio11 variable again appears to be important, contributing 2.1% to the distribution model. Although this is a smaller percentage compared to the other species, it still indicates that the mean temperature of the coldest quarter has some influence on the distribution of this species. The mean diurnal range (Bio2) has a contribution of 1.2%, which is relatively minor but still noteworthy. Overall, the bio variables related to temperature, such as the mean temperature of the coldest or wettest quarter and the mean diurnal range, appear to be important factors influencing the distribution of these species. For amphibians, environmental temperature has a significant impact on their behaviors; temperatures that are either too high or too low can affect amphibian behaviors such as predator avoidance, foraging, and reproductive activities [85,86]. Temperature not only affects species’ metabolic rates but also influences their growth, development, and reproductive cycles, especially during the breeding season. Suitable temperature ranges are closely related to the hatching of eggs, the growth and development rate of juveniles, and their survival rates [87]. *L. shihi* feeds on aquatic insects, shrimp, and algae [33], and temperature changes can affect the growth of plants and the activity of small organisms such as insects, thereby influencing this species’ distribution. The specific contribution of each variable varies among the three species observed, reflecting their unique ecological requirements and sensitivities to climatic conditions.

The availability of water sources significantly impacts the habitat quality for amphibians. Due to constraints imposed by topography and water availability, amphibian habitats are often limited in their extent [88,89]. Results from an environmental factor analysis indicate that the distance to the nearest surface water (DW, with a percent contribution of 58.2%) is crucial for the habitat quality of *Tylototriton wenxianensis*, likely related to the species’ breeding habits in still-water ponds [33]. When *T. wenxianensis* are located far from water sources, they must expend more energy to find suitable aquatic environments for breeding and living, which may reduce their breeding success and the survival rate of their offspring. Additionally, the proximity to water sources can significantly affect whether *T. wenxianensis* can access sufficient food resources. The areas surrounding water bodies typically harbor rich biodiversity, including various insects, small crustaceans, and worms, which are potential food sources for *T. wenxianensis*.

### 4.2. Key Factors Influencing the Predicted Distributional Shifts of Liua shihi, Pseudohynobius jinfo, and Tylototriton wenxianensis under Climate Change and Implications for Conservation

*Liua shihi*, *Pseudohynobius jinfo*, and *Tylototriton wenxianensis* have distinct habitat preferences. *L. shihi* primarily inhabits mountain streams ranging from mid-mountain to mountaintop areas, favoring an aquatic lifestyle; *P. jinfo* mainly resides in mountaintop marshes and slow-flowing streams and is predominantly aquatic; meanwhile, *T. wenxianensis* mainly occupies mountaintop marsh wetlands, leaning towards a terrestrial way of life [33,39]. Differences in these habitat preferences may be the primary reasons for the variations in their distribution prediction results. By analyzing the distribution prediction outcomes for these three caudate species, it was found that both natural and anthropogenic factors have impacted their distribution; yet, the main influencing factors differ significantly due to distinct habitat selection preferences. Therefore, in the process of conserving these species, it is imperative to consider the main influencing factors for each species and implement targeted conservation measures.

In the predictive distribution model for *Liua shihi*, the three most significant percentage contributions are the distance to the nearest road, the mean temperature of the coldest quarter, and isothermality. The environmental factor related to roads plays a predominant role in influencing the habitat of *L. shihi*, likely due to the broad altitudinal range of its distribution in the mountainous areas of Chongqing, spanning from mid to high elevations, which overlaps significantly with human activities. The mean temperature of the coldest quarter also exerts a substantial impact on the predictive distribution of *L. shihi*. This suggests that *L. shihi* possesses a considerable capacity for temperature adaptation, with a propensity for residing in habitats characterized by lower-temperature conditions. This preference is correlated with its natural habitat of high-altitude streams. Isothermality represents the third most significant environmental variable in the distribution prediction for *L. shihi*, accounting for a 5.5% influence. This index reflects the stability of seasonal temperature fluctuations. It is plausible that a stable pattern of seasonal temperature changes provides a consistent ecological environment, which *L. shihi* may preferentially select. In the process of conserving *L. shihi*, it is important to minimize human disturbances and maintain roadless areas within protected zones where the species is present. Concurrently, it is essential to manage the impacts of climate change and mitigate the occurrence of extreme weather events. Measures such as energy conservation and emission reduction can be adopted to decelerate the pace of climate change.

The three most significant predictors affecting the distribution range forecast of *Pseudohynobius jinfo*, as indicated by their percentage contributions, are the mean temperature of the wettest quarter, the distance to the nearest road, and the mean diurnal range. Natural factors are the primary influences on the predictive distribution of *P. jinfo*, which may be due to the low population density within the habitats located in the Chongqing municipality, resulting in a minimal spatial overlap with human activities. Among the natural factors, the mean temperature of the wettest quarter is identified as the most influential predictor for the distribution of *P. jinfo*. This could be attributed to the fact that the species inhabits mountainous regions at elevations ranging from 1700 to 2150 m, where adequate temperatures during the breeding season are crucial for the reproduction of the species. Roads also have a certain impact on the distribution of *P. jinfo*, indicating that the salamander’s habitat is subject to disturbances resulting from human activities. The mean diurnal range represents the difference between the daily maximum and minimum temperatures. A moderate variation in the temperature range can provide a greater array of ecological niches and adaptability, which is conducive to the reproduction and adaptation of species. In the conservation efforts for *P. jinfo*, it is essential to continue preserving these high-altitude habitats and minimize disturbances from roads.

In the study of the distribution of *Tylototriton wenxianensis*, the three most significant factors contributing to the model’s predictive accuracy were the distance to the nearest surface water, the distance to the nearest road, and elevation. Natural factors play a predominant role in shaping the predicted distribution of *T. wenxianensis*, which may be associated with its habitat being located at higher altitudes. The high percent contribution of the distance to the nearest surface water indicates a strong dependence of *T. wenxianensis* on aquatic environments, consistent with the characteristic life stage of its larvae, which reside in isolated, high-altitude still-water pools. Roads also exert a significant impact on the distribution of *T. wenxianensis*. The construction of roads and the associated human activities can lead to habitat destruction, disturbance, and fragmentation, thereby exerting a detrimental effect on the survival and migration of *T. wenxianensis*. Consequently, the greater the distance from the nearest road, the more expansive the distribution of *T. wenxianensis* is likely to be, suggesting better integrity and connectivity of its habitat. Elevation is another critical factor that can influence the distribution of *T. wenxianensis*. As *T. wenxianensis* typically inhabits higher altitudinal zones (900–1400 m) [39], elevation directly affects the environmental conditions necessary for this species’ survival and reproduction. Therefore, in the conservation endeavors for *T. wenxianensis*, it is imperative to safeguard this species’ dispersed breeding ponds and ensure their adequate provision. During the planning and construction of roads, efforts should be made to circumvent or minimize the disruption of *T. wenxianensis* habitats.

### 4.3. Changes in the Predicted Distribution Areas of Liua shihi, Pseudohynobius jinfo, and Tylototriton wenxianensis in the Mountainous Regions of Chongqing under Climate Change Conditions

Under the conditions of climate warming, amphibians often migrate to higher altitudes and latitudes [90]. Research by Duan et al. (2016) found that approximately 90% of amphibian species in China are predicted to migrate northward in the future. However, in our study, aside from *Liua shihi*, which is expected to move northward from 2050 to 2090, *Pseudohynobius jinfo* is projected to shift southeastward, and *Tylototriton wenxianensis* is anticipated to move southeastward. This phenomenon may be partly due to the unique climatic characteristics of Chongqing, where the Qinling–Daba Mountains to the north act as a barrier against the incursion of cold northern air, coupled with the region’s significant topographical relief and uneven vegetation distribution. Additionally, the presence of complex local microclimates in Chongqing may also be related to the species’ specific locations and their particular physiological tolerances [91]. It is noteworthy that caudate amphibians generally have limited migratory capabilities. The habitats of these three species are particularly narrow, with *P. jinfo* and *T. wenxianensis* being prime examples. Furthermore, the complex terrain of Chongqing could make them more susceptible to the impacts of climate change, posing risks to their survival and even threatening them with extinction.

## 5. Conclusions

Under the dual influences of climate change and human activities, this study predicts significant alterations in the suitable habitats for three Caudata species: *Liua shihi*, *Pseudohynobius jinfo,* and *Tylototriton wenxianensis*. From the present moment until the year 2050, the suitable habitat for *L. shihi* is expected to expand, while the habitats for *P. jinfo* and *T. wenxianensis* are projected to contract. By 2070 and 2090, the suitable habitats for all three species are anticipated to further diminish.

Among the various factors influencing the predicted changes in the distribution areas of the three caudate amphibians in Chongqing, human disturbance, particularly the distance to the nearest road, has the most significant impact on *Liua shihi*. Climate variables and human disturbance, including the mean temperature of the wettest quarter, the mean diurnal range, and the distance to the nearest road, are the primary factors affecting *Pseudohynobius jinfo*. For *Tylototriton wenxianensis*, habitat variables, especially the distance to water sources, are the most critical influencing factors.

Future conservation strategies for *Liua shihi*, *Pseudohynobius jinfo*, and *Tylototriton wenxianensis* in Chongqing should particularly focus on the following key areas: the Chengkou County, the Wuxi County, the Wushan County, the Yunyang County, the Fengjie County, and the Kaizhou District in the northeastern region are critical conservation areas for *L. shihi*; the southern regions of the Nanchuan District and the Wulong District serve as key conservation areas for *P. jinfo*; meanwhile, the Yunyang County and the Fengjie County in the northeastern region are vital for the protection of *T. wenxianensis*.

In the implementation of conservation activities, targeted protective measures should be adopted for different species, including the following: (1) *Liua shihi* is distributed in the northeastern part of Chongqing. Its range encompasses several nature reserves, including the Daba Mountains National Nature Reserve in Chengkou, the Wulipo National Nature Reserve in Wushan, the Yintiaoling National Nature Reserve in Wuxi, and the Hongchiba National Forest Park, among others. Additionally, this species can be found in small mountain streams outside of these protected areas. For the protection of *L. shihi*, it is essential to minimize human disturbance. In protected areas where this species is found, regions without roads should be maintained, and the vegetation within the distribution area should be carefully preserved. (2) The distribution range of *Pseudohynobius jinfo* is narrow, primarily occurring within the confines of the Jinfo Mountain National Nature Reserve in Nanchuan and the Baima Mountain Municipal Nature Reserve in Wulong. In the conservation of *P. jinfo*, it is important to continue preserving its high-altitude habitats and reduce road disturbances. (3) *Tylototriton wenxianensis* is predominantly located outside designated conservation areas in Chongqing. The preservation of this species necessitates the protection and maintenance of both the quality and quantity of its aquatic habitats due to its strong reliance on water bodies. Additionally, it is of paramount importance to address the detrimental effects of road construction on the survival and migration patterns of *T. wenxianensis* by ensuring that the integrity and connectivity of their habitats are maintained. It is proposed that a specialized small-scale conservation area be established within the suitable distribution range of *T. wenxianensis*, with a focus on this species as the primary conservation target. Moreover, given the proximity of *T. wenxianensis* habitats to human settlements, it is vital to foster public awareness and understanding of the significance of conserving this species through educational initiatives. Such efforts are expected to reduce the negative impacts on the habitats of these three salamander species.

## Figures and Tables

**Figure 1 animals-14-00672-f001:**
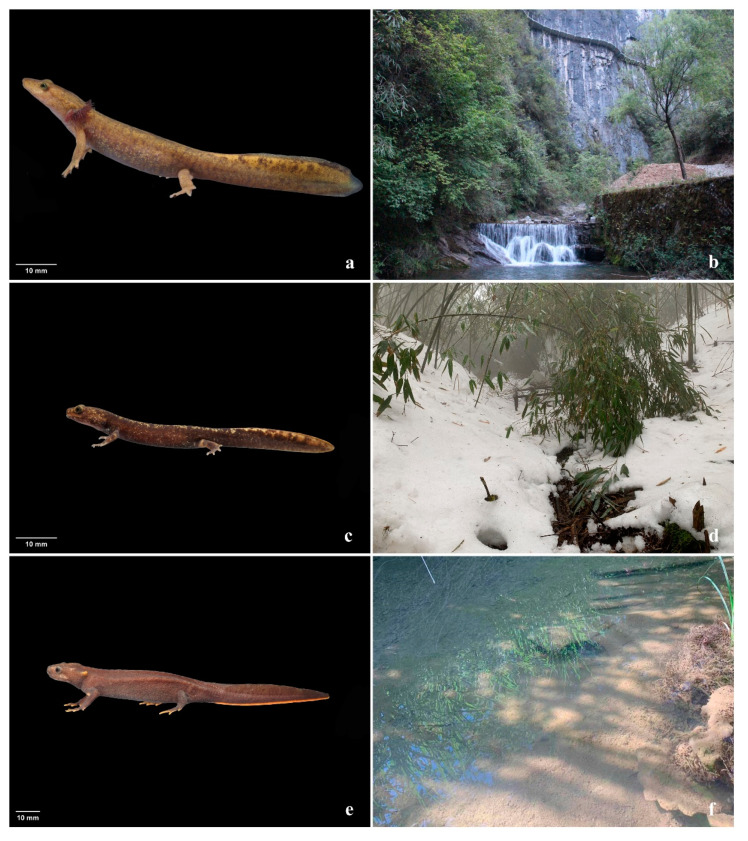
*Liua shihi* (**a**) and its natural habitat (**b**), *Pseudohynobius jinfo* (**c**) and its natural habitat (**d**), and *Tylototriton wenxianensis* (**e**) and its natural habitat (**f**). Photos taken by Qi Ma.

**Figure 2 animals-14-00672-f002:**
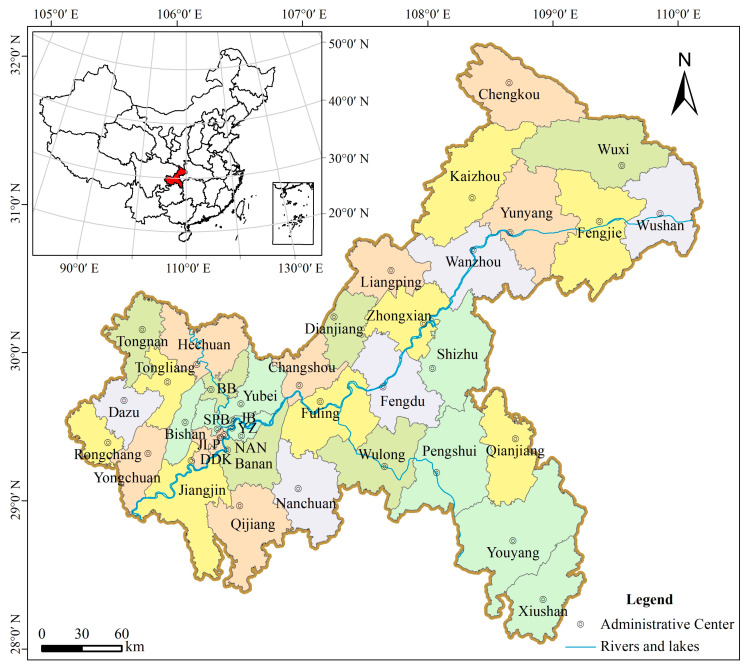
Map of the Chongqing municipality. The abbreviations BB, SPB, JB, DDK, NAN, and JLP, respectively, correspond to the Beibei District, the Shapingba District, the Jiangbei District, the Dadukou District, the Nan’an District, and the Jiulongpo District.

**Figure 3 animals-14-00672-f003:**
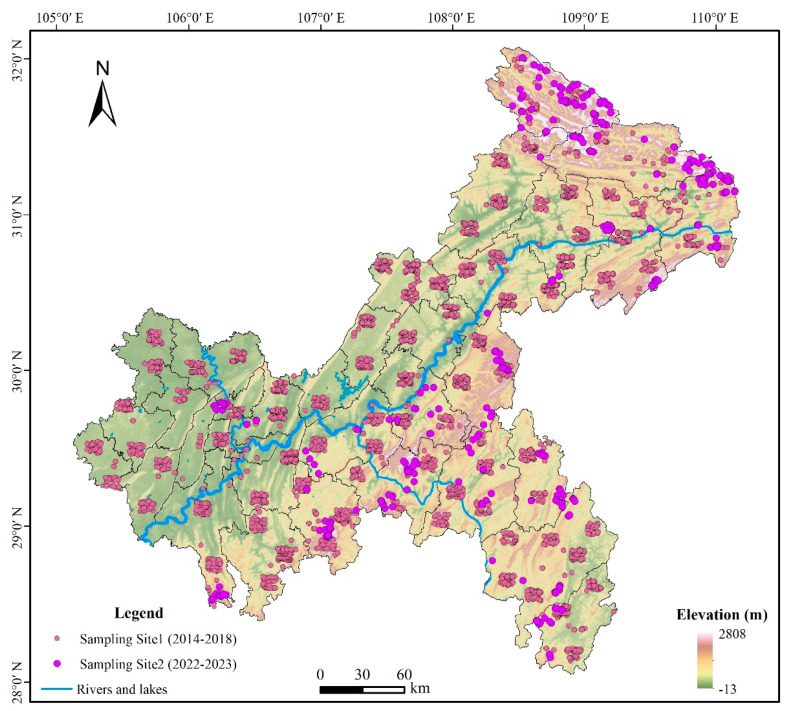
Distribution map of amphibian survey transects in Chongqing. Sampling Site 1 (2014–2018) refers to the Second National Survey of Terrestrial Wildlife Resources in Chongqing, while Sampling Site 2 (2022–2023) denotes specialized surveys specifically targeting salamanders.

**Figure 4 animals-14-00672-f004:**
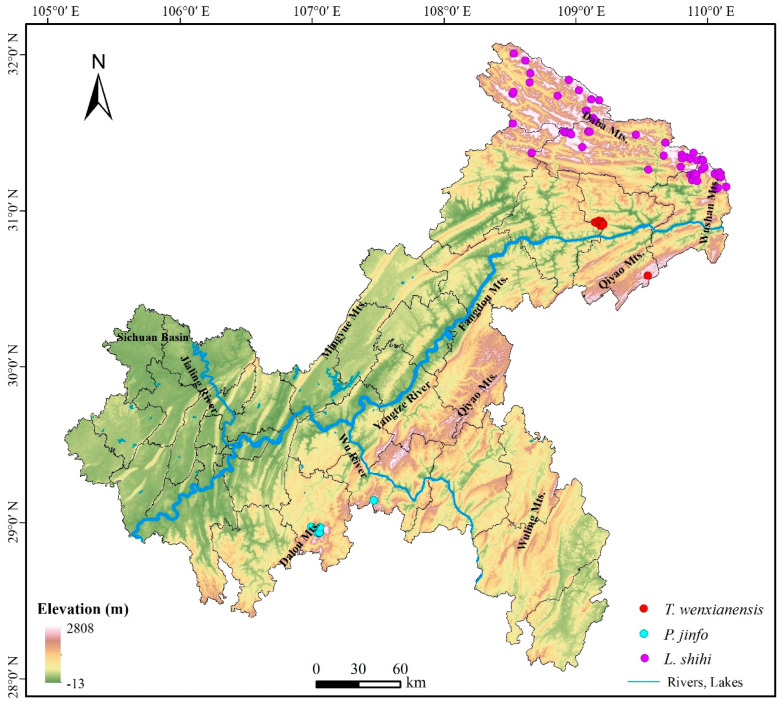
Distribution of valid occurrence sites for *Liua shihi*, *Pseudohynobius jinfo*, and *Tylototriton wenxianensis* in after deduplication in Chongqing.

**Figure 5 animals-14-00672-f005:**
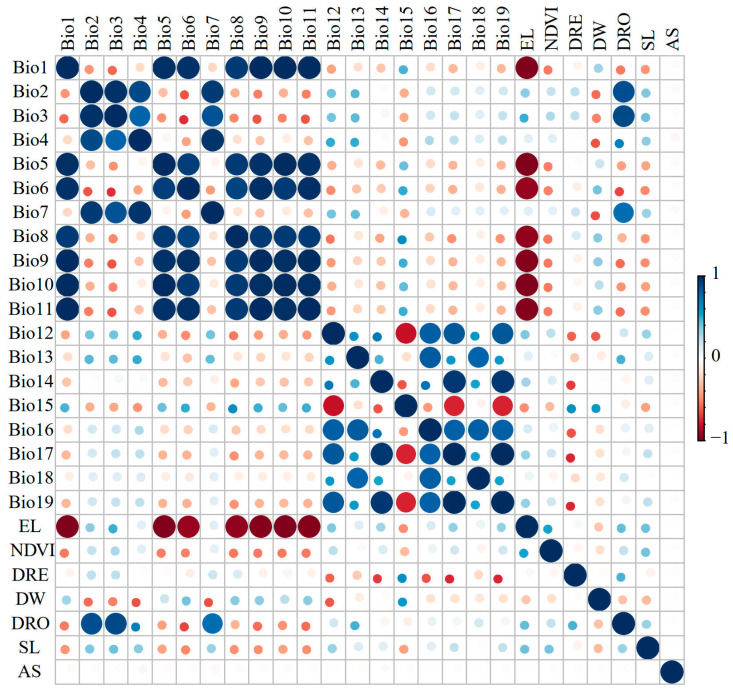
Correlation coefficients’ matrix among 26 environmental variables. Positive correlations are displayed in blue and negative correlations in a red color. The color intensity and the size of the circle are proportional to the correlation coefficients.

**Figure 6 animals-14-00672-f006:**
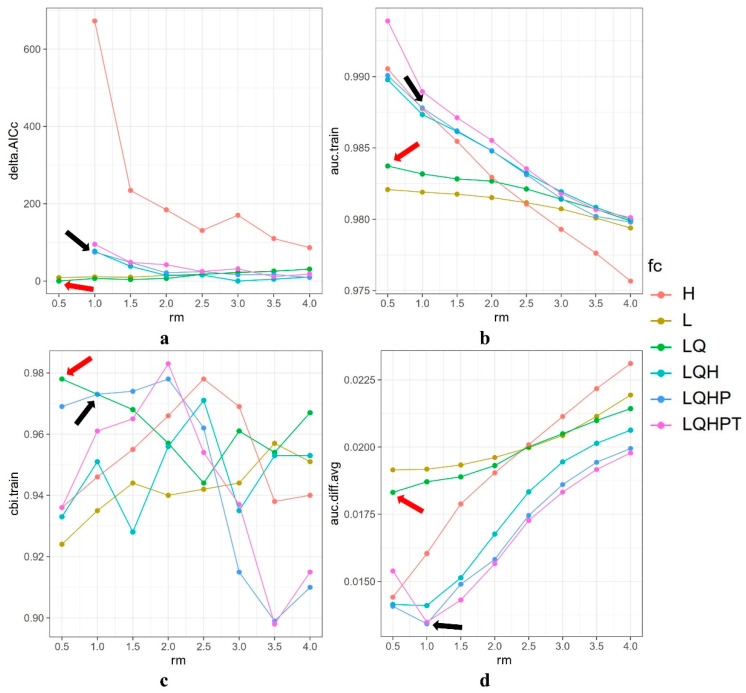
Optimization results for the MaxEnt mode of *Liua shihi* under different parameter settings, (**a**) delta.AICc, (**b**) AUC.train, (**c**) CBI.train, and (**d**) AUC.diff. Feature classes (H, hinge; L, linear; LQ, linear quadratic; LQH, linear quadratic hinge; LQHP, linear quadratic hinge product; and LQHPT, linear quadratic hinge product threshold). Black arrow indicates the default setting, and the red arrow indicates the AICc-chosen setting.

**Figure 7 animals-14-00672-f007:**
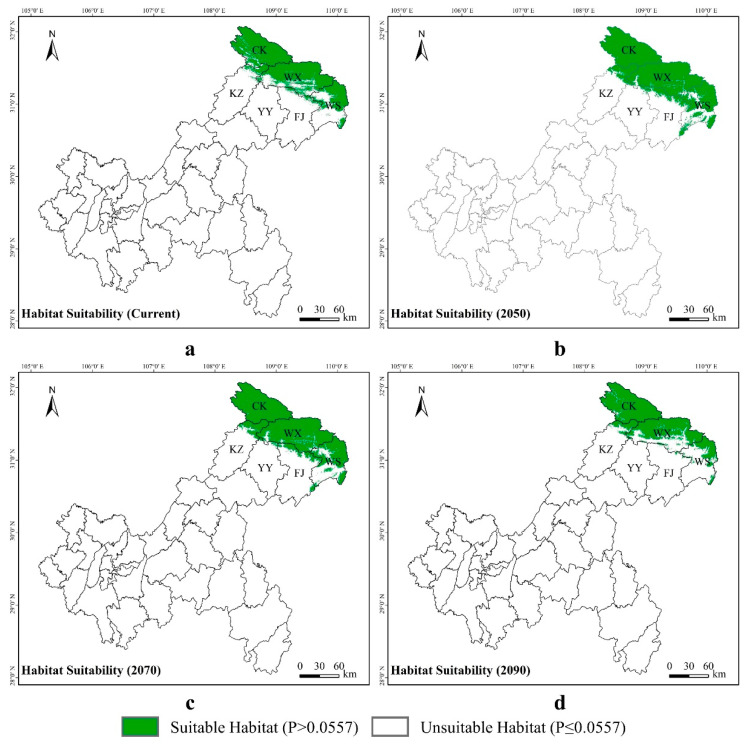
Potential distribution of *Liua shihi* predicted by the MaxEnt model in Chongqing under different climate scenarios: (**a**) current climate, (**b**) 2050, (**c**) 2070, and (**d**) 2090. The abbreviations CK, WX, WS, FJ, KZ, and YY, respectively, correspond to Chengkou County, Wuxi County, Wushan County, Fengjie County, Kaizhou District, and Yunyang County within the Chongqing municipality, with the same abbreviations used below.

**Figure 8 animals-14-00672-f008:**
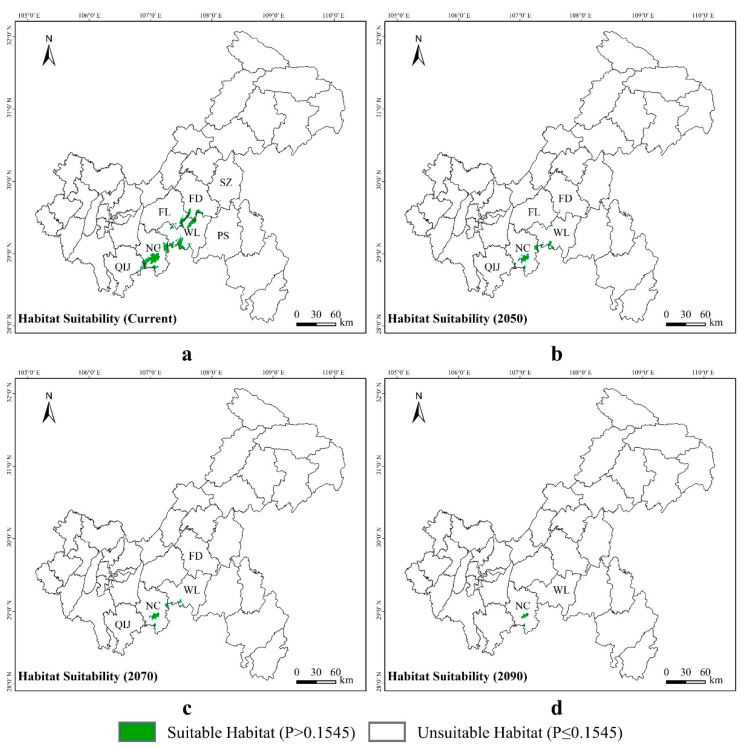
Potential distribution of *Pseudohynobius jinfo* predicted by the MaxEnt model in Chongqing under different climate scenarios: (**a**) current climate, (**b**) 2050, (**c**) 2070, and (**d**) 2090. The abbreviations NC, WL, PS, FL, FD, QIJ, and SZ, respectively, correspond to Nanchuan District, Wulong District, Pengshui County, Fuling District, Fengdu County, Qijiang District, and Shizhu County within the municipality of Chongqing, with the same abbreviations used below.

**Figure 9 animals-14-00672-f009:**
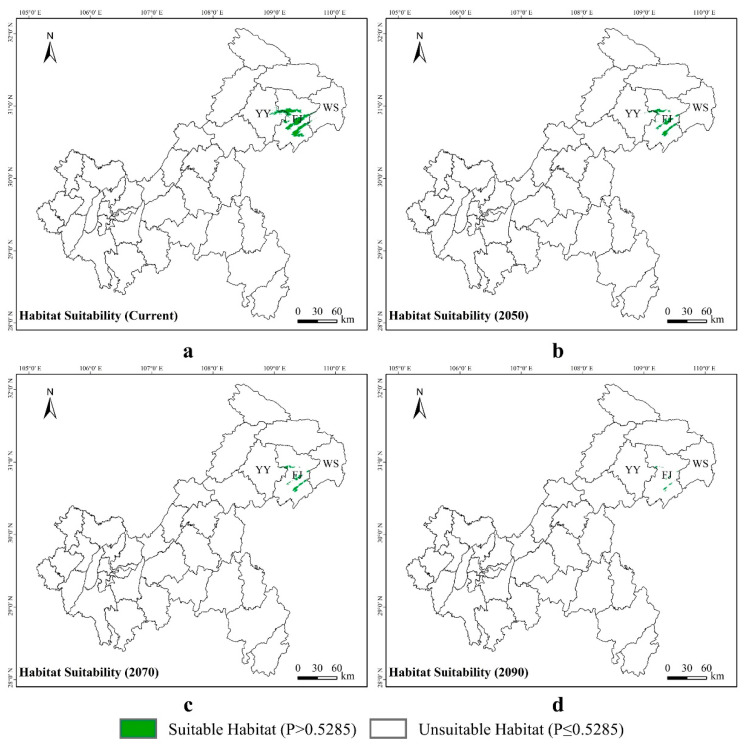
Potential distribution of *Tylototriton wenxianensis* predicted by the MaxEnt model in Chongqing under different climate scenarios: (**a**) current climate, (**b**) 2050, (**c**) 2070, and (**d**) 2090. The abbreviations FJ, YY, and WS, respectively, correspond to Fengjie County, Yunyang County, and Wushan County within the municipality of Chongqing, with the same abbreviations used below.

**Figure 10 animals-14-00672-f010:**
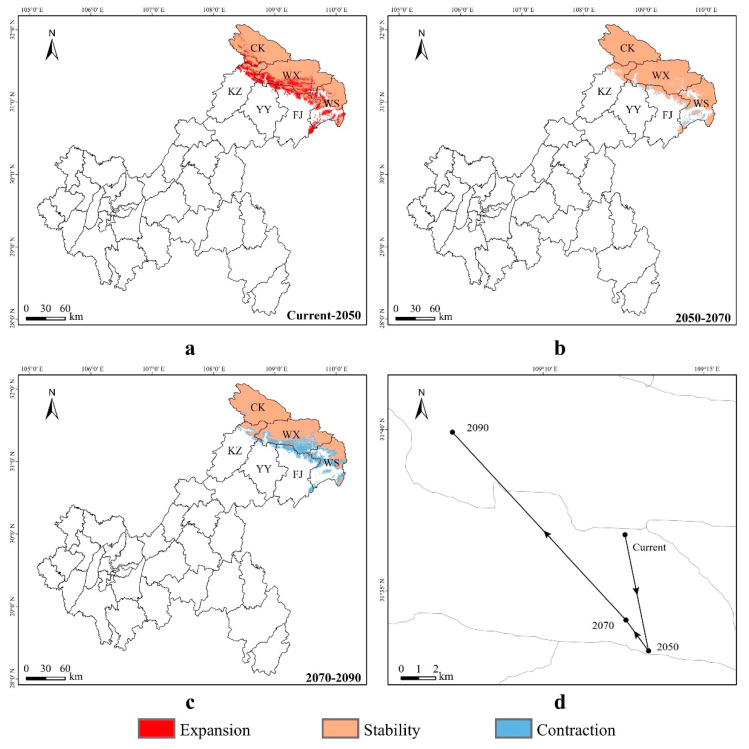
Changes in the potential geographic range of *Liua shihi* under different climate scenarios: (**a**) current–2050, (**b**) 2050–2070, (**c**) 2070–2090, and (**d**) centroid transfer trajectory. The black arrow lines depicted in Figure d represent the centroid transfer routes, with the arrows indicating the directions of transfer.

**Figure 11 animals-14-00672-f011:**
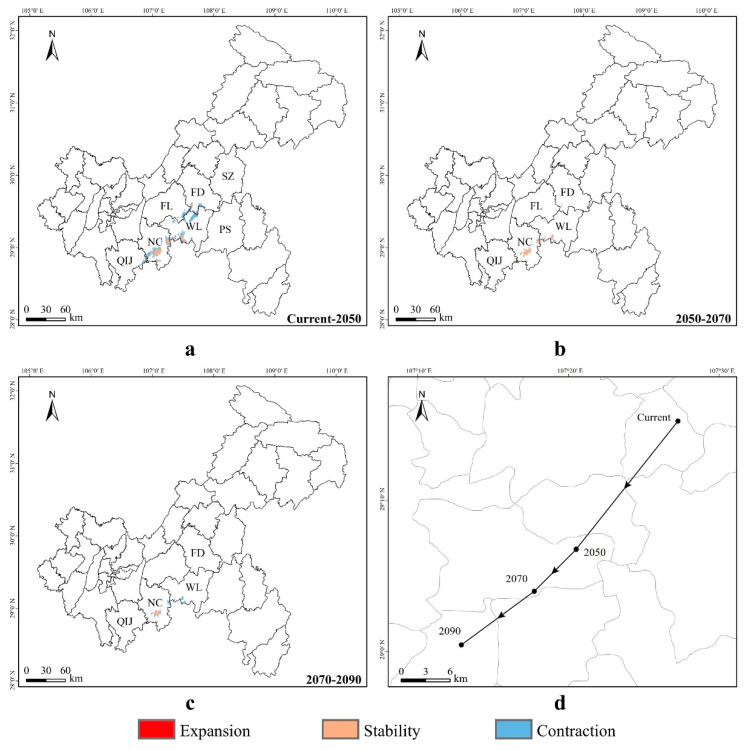
Changes in the potential geographic range of *Pseudohynobius jinfo* under different climate scenarios: (**a**) current–2050; (**b**) 2050–2070; (**c**) 2070–2090; and (**d**) centroid transfer trajectory. The black arrow lines depicted in Figure d represent the centroid transfer routes, with the arrows indicating the directions of transfer.

**Figure 12 animals-14-00672-f012:**
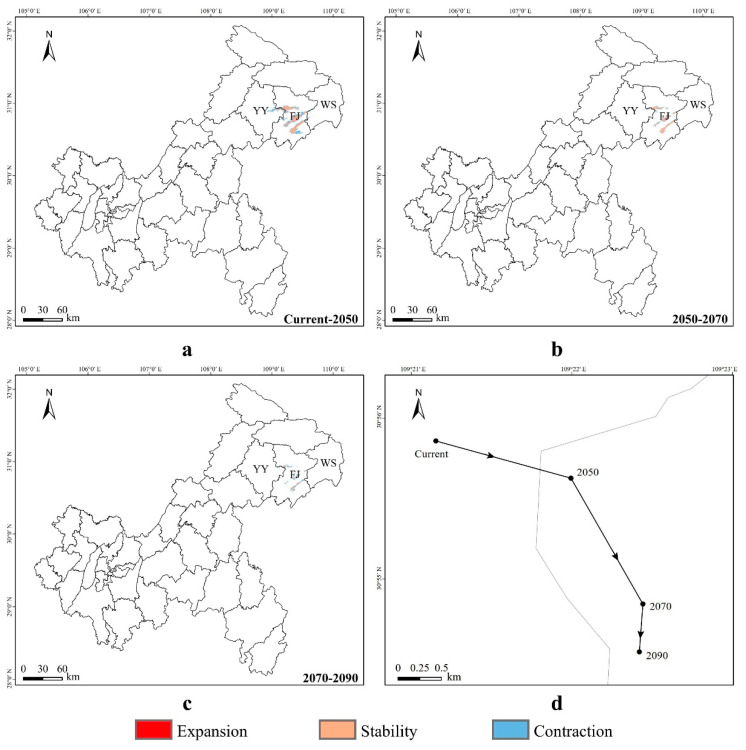
Changes in the potential geographic range of *Tylototriton wenxianensis* under different climate scenarios: (**a**) current–2050; (**b**) 2050–2070; (**c**) 2070–2090; and (**d**) centroid transfer trajectory. The black arrow lines depicted in Figure d represent the centroid transfer routes, with the arrows indicating the directions of transfer.

**Figure 13 animals-14-00672-f013:**
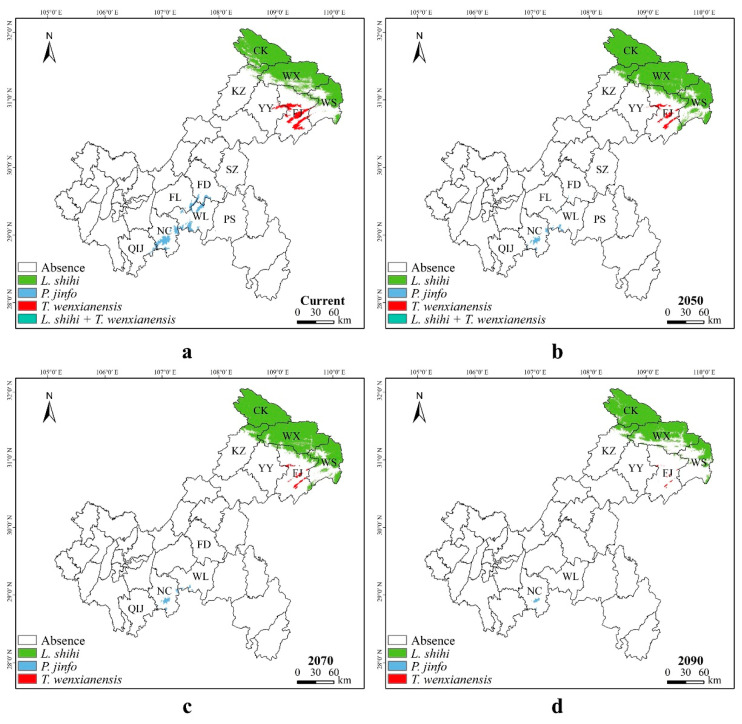
Overlapping suitable area prediction for *Liua shihi*, *Pseudohynobius jinfo*, and *Tylototriton wenxianensis* in Chongqing during the (**a**) present, (**b**) 2050, (**c**) 2070, and (**d**) 2090 stages.

**Table 1 animals-14-00672-t001:** Environment variables for MaxEnt modeling.

Environment Variable	Variable Type
Bio1	Annual Mean Temperature/°C
Bio2	Mean Diurnal Range/°C
Bio3	Isothermality [(BIO2/BIO7) × 100]
Bio4	Temperature Seasonality
Bio5	Max Temperature of Warmest Month/°C
Bio6	Min Temperature of Coldest Month/°C
Bio7	Temperature Annual Range (BIO5-BIO6)/°C
Bio8	Mean Temperature of Wettest Quarter/°C
Bio9	Mean Temperature of Driest Quarter/°C
Bio10	Mean Temperature of Warmest Quarter/°C
Bio11	Mean Temperature of Coldest Quarter/°C
Bio12	Annual Precipitation/mm
Bio13	Precipitation of Wettest Month/mm
Bio14	Precipitation of Driest Month/mm
Bio15	Precipitation Seasonality (Coefficient of Variation)
Bio16	Precipitation of Wettest Quarter/mm
Bio17	Precipitation of Driest Quarter/mm
Bio18	Precipitation of Warmest Quarter/mm
Bio19	Precipitation of Coldest Quarter/mm
AS	Aspect
SL	Slope
EL	Elevation/m
DRO	Distance to the nearest road/m
DRE	Distance to the nearest residential area/m
DW	Distance to the nearest surface water/m
NDVI	Normalized Difference Vegetation Index

## Data Availability

The data used in the analysis can be obtained from the authors upon request.

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
