# Peer review of "Impact of Climate Change on the Distribution of Three Rare Salamanders (Liua shihi, Pseudohynobius jinfo, and Tylototriton wenxianensis) in Chongqing, China, and Their Conservation Implications"

_animals, 2024, doi:10.3390/ani14050672_

Round 1

Reviewer 1 Report

Comments and Suggestions for Authors

This is a good study. Certainly information on Chinese salamander species is important, and I applaud the effort to maximize the returns on what information can be gathered on these three species.

The annotated Word copy of your manuscript accompanying this review has a number of marginal comments and some suggested changes to the text (these last are mostly rephrasing things as a native English speaker would – your written English is generally quite clear and I had no trouble understanding the text). I hope that you find these useful.

Please consider the issues raised in the marginal comments. These are mostly relatively minor issues but do require attention (i.e. distinguishing between Chongqing the city and Chongqing the province).

General points:       

The Introduction is good, but a couple of references on the natural histories of each of these species would be helpful. Most of us outside of East Asia will not be familiar with them.

A review of historical data on each of these species is necessary here – have their distributions changed over the past few decades? What are historical population size trends? What are their ranges outside of Chongqing? If these questions can’t be answered from the data available to you, say so – it’s valuable to know if a species is poorly-known.

A more detailed description of the vegetation communities occupied by each of these species would be informative – I realize that you used an index in your models but such things as percent canopy cover and percent ground cover are important for salamanders.

In the Methods, we need to be told more about your field surveys – were you just collecting presence/absence data, or were you obtaining some estimate of salamander numbers at each site? Also, what was the criterion for judging a site lacking in salamanders? If you collected any data on numbers of salamanders at any given site, these data should be presented here – they will give the reader a better idea of population sizes, and their variance over the geographic ranges of these species may be informative for conservation purposes.

I have no other general criticisms to make – the marginal comments on the Word copy of the ms cover everything else, I think.

This is a good, innovative study and will be a valuable resource for conservation efforts concerning these species. I hope that these salamanders benefit from your work. Good luck with your further endeavours.

Comments on the Quality of English Language

The written English is generally commendably clear and I had little trouble understanding what the authors had to say. I have made a few suggestions and notes in the annotated version of the ms included with this review - most of these involve ways that a native English speaker would phrase things, or query what was actually meant in cases where I couldn't understand what the authors were saying. There aren't many of these, however.

Author Response

We greatly appreciate the time and effort you have dedicated to reviewing our manuscript. We have revised the manuscript in accordance with your suggestions and comments. Specifically, we have improved our title, expanded the introduction to include more on the natural history and distribution of the three salamanders, rechecked and reanalyzed our data, and provided a more detailed description of our methods. Additionally, we have updated our results and discussion sections, redrawn the figures for enhanced readability, and refined the language to reduce repetition (our responses are highlighted in red).

Below are our responses to your comments:

Comments 1: This is a good study. Certainly information on Chinese salamander species is important, and I applaud the effort to maximize the returns on what information can be gathered on these three species.

Response 1: Thank you for your nice comment. We are glad this research could provide reference for the conservation of these small and less-known unremarkable salamanders.

Comments 2: The annotated Word copy of your manuscript accompanying this review has a number of marginal comments and some suggested changes to the text (these last are mostly rephrasing things as a native English speaker would – your written English is generally quite clear and I had no trouble understanding the text). I hope that you find these useful.

Response 2: Thank you very much! We found your marginal comments very helpful, and we have revised the manuscript according to your comments and suggestions, please check the replies to your comments in the attached pdf file.

Comments 3:Please consider the issues raised in the marginal comments. These are mostly relatively minor issues but do require attention (i.e. distinguishing between Chongqing the city and Chongqing the province).

Response 3: We found your marginal comments very helpful, and we have revised the manuscript according to your comments and suggestions. The administrative levels of places have been stated more clearly in this version of the manuscript.

Comments 4: The Introduction is good, but a couple of references on the natural histories of each of these species would be helpful. Most of us outside of East Asia will not be familiar with them.

Response 4: Thank you, we have expanded the descriptions of the natural histories of these species and have provided the appropriate references.

Comments 5: A review of historical data on each of these species is necessary here – have their distributions changed over the past few decades? What are historical population size trends? What are their ranges outside of Chongqing? If these questions can’t be answered from the data available to you, say so – it’s valuable to know if a species is poorly-known.

Response 5: Thank you! We have expanded descriptions of the natural histories of these species. We have provided descriptions of the population trends of these animals, with references. However, historical surveys on the distribution areas of these species are not extensive and often do not have precise location records (with longitude and latitude) due to limited technology back in time. Therefore, we could not assess how their distributions have changed over the past few decades. We have supplemented information about their ranges outside of Chongqing in the revised introduction.

Comments 6: A more detailed description of the vegetation communities occupied by each of these species would be informative – I realize that you used an index in your models but such things as percent canopy cover and percent ground cover are important for salamanders.

Response 6: Thank you! We have added a detailed description of the vegetation communities in the introduction.

Comments 7: In the Methods, we need to be told more about your field surveys – were you just collecting presence/absence data, or were you obtaining some estimate of salamander numbers at each site? Also, what was the criterion for judging a site lacking in salamanders? If you collected any data on numbers of salamanders at any given site, these data should be presented here – they will give the reader a better idea of population sizes, and their variance over the geographic ranges of these species may be informative for conservation purposes.

I have no other general criticisms to make – the marginal comments on the Word copy of the ms cover everything else, I think.

Response 7: Thank you! We have revised the Methods section and addressed your concerns in the updated manuscript. The data were primarily derived from field surveys, which we have now clearly stated. We have also provided details on how the field surveys were conducted.

In addition, we have re-examined our methods and datasets manually. We have found few anomalous distribution records based on museums' collection (one abnormal distribution site for Pseudohynobius jinfo and Tylototriton wenxianensis, two abnormal distribution sites for Liua shihi). These abnormal sites were identified by their elevation being much lower than elevation records of relevant references and field surveys. This issue may have been caused by an inaccurate description of occurrence sites of some museums' collection. We excluded these abnormal sites and reanalyzed our revised dataset. Results indicate that the potential distribution areass of P. jinfo and T. wenxianensis are much smaller than those in our last manuscript. However, this is more reasonable since these species are distributed in high elevation area in Chongqing, which is only a small portion of the total area. The updated potential distribution areas of the three salamanders are also more consistent with our knowledge of their natural history and habitat based on both field survey and relevant references. We are lucky that we have found this flaw before it might mislead the conservation on these animals, and we deeply appreciate your help.

To address the question of how we determined the areas where the three salamanders are not distributed in Chongqing, we added data from additional extensive field work relevant to amphibians in Chongqing from July 2014 to July 2018. We have extensively surveyed entire Chongqing covering 1760 transects, and the absence of the three salamanders in the lowlands of Chongqing were confirmed through field surveys.

Comments 8: This is a good, innovative study and will be a valuable resource for conservation efforts concerning these species. I hope that these salamanders benefit from your work. Good luck with your further endeavours.

Response 8: Thank you for your encouragement. We are glad our research could help the conservation of these little-known salamanders.

Comments 9: The written English is generally commendably clear and I had little trouble understanding what the authors had to say. I have made a few suggestions and notes in the annotated version of the ms included with this review - most of these involve ways that a native English speaker would phrase things, or query what was actually meant in cases where I couldn't understand what the authors were saying. There aren't many of these, however.

Response 9: Thank you! It is very nice of you to provide helpful suggestions and notes in the annotated version of the manuscript included with your review. We have revised them; please check the attached PDF file.

Reviewer 2 Report

Comments and Suggestions for Authors

The paper is generally well written and clearly presented. I am not qualified to comment on all the methods, but do not see any outstanding issues apart from several questions written below.

In the 'simple summary', the term 'umbrella species' (line 19) needs to be defined, as this summary is presumably for general rather than specialised readers.

Line 251: 'MaxEnt has demonstrated a high predictive performance'. References are need for this statement, to be placed immediately after the statement. It would also be appropriate to include a brief description of what the MaxEnt model is designed to do, and how it works.

line 358 and following: I do not see any references as to how the predictions for climate scenarios in 2050, 2070, 2090 have been reached. How were these predictions obtained?

Author Response

We greatly appreciate your dedication of time and effort in reviewing this manuscript. Thank you for your nice comments. We have revised our manuscript based on your suggestions and comments. Generally, we improved our title; expanded our introduction on the natural history and distribution of the three salamanders; rechecked our data and reanalyzed it, supplemented detailed description on methods; updated our results and discussion; figures were redrawn to make it more readable; language was also improved to reduce repetition (our responses are all red-colored).

1. We have re-examined our methods and datasets manually. We have found few abnormal distribution records based on museum collections (one abnormal distribution site for Pseudohynobius jinfo and Tylototriton wenxianensis, two abnormal distribution sites of Liua shihi). These abnormal sites were identified by their elevation being significantly lower than elevation records of relevant references and field surveys. This may be due to inaccurate description of occurrence sites of some museum collections. We excluded these abnormal sites and reanalyzed our revised dataset. The results indicate that the potential distribution areas of P. jinfo and T. wenxianensis are much smaller than those reported in our previous manuscript. However, this is more reasonable since these species are distributed in the high-elevation areas of Chongqing, which constitutes only a small portion of the area. The updated potential distribution areas of the three salamanders are also more consistent with our knowledge of their natural history and habitat based on both field survey and relevant references.

2. To address the question of how we determined the areas where the three salamanders do not distribute in Chongqing, we added data from additional extensive field work relevant to amphibians in Chongqing from July 2014 to July 2018. We extensively surveyed the entire Chongqing region covering 1760 transects, and the absence of the three salamanders in the lowlands of Chongqing has been confirmed by field surveys.

3. The introduction to natural histories and distribution of the three salamanders was expanded and relevant references were provided.

4. Figures have been updated and redrawn. We have added a map showing the administrative divisions of Chongqing Municipality.

Following are the replies to your comments:

Comments 1: The paper is generally well written and clearly presented. I am not qualified to comment on all the methods, but do not see any outstanding issues apart from several questions written below.

Response 1: Thank you for your valuable feedback.

Comments 2: In the 'simple summary', the term 'umbrella species' (line 19) needs to be defined, as this summary is presumably for general rather than specialised readers.

Response 2: Thank you, we have added a definition for it in the simple summary.

Comments 3: Line 251: 'MaxEnt has demonstrated a high predictive performance'. References are need for this statement, to be placed immediately after the statement. It would also be appropriate to include a brief description of what the MaxEnt model is designed to do, and how it works.

Response 3: Thank you for your valuable feedback. We have now included the necessary references immediately following the statement regarding MaxEnt's high predictive performance. Additionally, we have incorporated a concise description of the design objectives and operational principles of the MaxEnt model within the text, specifically in lines 370-391.

Comments 4: line 358 and following: I do not see any references as to how the predictions for climate scenarios in 2050, 2070, 2090 have been reached. How were these predictions obtained?

Response 4: Thank you very much for your comment. In response to your suggestion, we have made the appropriate revisions in the manuscript. For details on the methodology used to obtain the predictions for climate scenarios in 2050, 2070, and 2090, please refer to lines 328-332 and 380-385 of the manuscript.

Reviewer 3 Report

Comments and Suggestions for Authors

Animals-2805679: Potential distribution of three endemic rare salamanders in Chongqing, China under Climate Change and Their Conservation Implications

Dear authors,

After reviewing the manuscript, I realized that the main text is well written and organized, and both figures and tables have good quality. The research design fits the objectives of the manuscript. The authors provided a good data and valuable discussion of the conservation implications of three extant salamanders from China that are threatened by climate change. Despite this, the co-authors should carry out some minor stylistic changes, all of them marked in the PDF file.

All the best.

Comments on the Quality of English Language

The quality of Language is good, and only minor editing is required. 

Author Response

We would like to express our sincere gratitude for the time and effort you have invested in reviewing our manuscript. Your constructive comments have been well received. In response, we have made several revisions to our manuscript. Specifically, we have refined the title, broadened the introduction to include more information on the natural history and distribution of the three salamander species, re-examined our data and conducted a re-analysis, and provided a more comprehensive description of our methods. Additionally, we have updated our results and discussion sections, and the figures have been redesigned for enhanced clarity. We have also made language improvements to minimize repetition. Please note that all of our responses are highlighted in red.

1. We have re-examined our methods and datasets manually. We have found few abnormal distribution records based on museum collections (one abnormal distribution site for Pseudohynobius jinfo and Tylototriton wenxianensis, two abnormal distribution sites of Liua shihi). These abnormal sites were identified by their elevation being much lower than elevation records of relevant references and field surveys. This may be due to inaccurate description of occurrence sites of some museum collections. We excluded these abnormal sites and reanalyzed our revised dataset. Results indicate that the potential distribution areas of P. jinfo and T. wenxianensis are much smaller than those in our previous manuscript. However, this is more reasonable since these species are distributed in high elevation areas in Chongqing, which is only a small portion of the area. The updated potential distribution areas of the three salamanders are also more consistent with our knowledge of their natural history and habitat based on both field survey and relevant references.

2. To address the question of how we determined the areas where the three salamanders do not distribute in Chongqing, we added data from additional extensive field work relevant to amphibians in Chongqing from July 2014 to July 2018. We have extensively surveyed entire Chongqing covering 1760 transects, and the absence of the three salamanders in the lowlands of Chongqing was confirmed by field surveys.

3. The introduction to natural histories and distribution of the three salamanders was expanded and relevant references were provided.

4. Figures were updated and redrawn. We added a map showing the administrative divisions of Chongqing Municipality.

Following are the replies to your comments:

Comments 1: After reviewing the manuscript, I realized that the main text is well written and organized, and both figures and tables have good quality. The research design fits the objectives of the manuscript. The authors provided a good data and valuable discussion of the conservation implications of three extant salamanders from China that are threatened by climate change. Despite this, the co-authors should carry out some minor stylistic changes, all of them marked in the PDF file.

Response 1: Thank you for your nice comments. We have revised the manuscript as you suggested. Please check the detailed responses in the attached pdf file.

Comments 2:The quality of Language is good, and only minor editing is required.

Response 2: Thank you! We have made improvements to our language this time around.

Reviewer 4 Report

Comments and Suggestions for Authors

Field surveys of endangered salamanders and predicting their population trends in the context of global climate change and other anthropogenic factors are very important for conservation measurements. My biggest concern is the incomplete methods. What did you do in the field? What did you measure or collect? Salamander specimens, temperature? Incomplete methods do not allow other researchers to replicate or check your research.  Some images need to improve, either too small legends or symbols that lack contrast and are difficult to see. The title of your manuscript should be shortened and improved, and the scientific names in parentheses should be included. Avoid repetitions in your text. Too often, do you mention rare and endemic. I have no idea what Class I and II means and the more commonly known IUCN threat status should be used. For further comments and corrections see my comments in the attached PDF.

Author Response

We are deeply grateful for the time and effort you have invested in reviewing our manuscript, and we appreciate the constructive feedback you have provided. In response to your comments, we have made substantial revisions to our manuscript. Specifically, we have refined the title, broadened the introduction to encompass the natural history and distribution of the three salamander species, meticulously rechecked and reanalyzed our data, and provided a more comprehensive description of our methods. Additionally, we have updated the results and discussion sections. The figures have been redesigned to enhance their clarity and readability. We have also revised the language throughout the manuscript to eliminate redundancy. Please note that all of our responses are indicated in red text.

To address the question of how we determined the areas where the three salamanders do not distribute in Chongqing, we added data from additional extensive field work relevant to amphibians in Chongqing from July 2014 to July 2018. We have conducted extensive surveys across the entirety of Chongqing, covering 1760 transects , and field surveys confirmed the absence of the three salamanders in the lowlands of Chongqing.

1. The introduction section on the natural history and distribution of the three salamander species has been expanded, and pertinent references have been included.

2. The figures have been updated and redrawn for clarity. Additionally, a map depicting the administrative divisions of Chongqing Municipality has been included.

Below are the responses to your comments:

Comments 1: Field surveys of endangered salamanders and predicting their population trends in the context of global climate change and other anthropogenic factors are very important for conservation measurements.

Response 1: I concur. Conducting field surveys of endangered salamanders and predicting their population trends in the context of global climate change and other anthropogenic factors are pivotal components of our research. These efforts are instrumental in informing conservation strategies for these vulnerable species.

Comments 2: My biggest concern is the incomplete methods. What did you do in the field? What did you measure or collect? Salamander specimens, temperature? Incomplete methods do not allow other researchers to replicate or check your research.

Response 2: We deeply appreciate this comment. We added details for Methods and answered these questions in the revised manuscript.

We re-examined our methods and datasets manually. We have found few abnormal distribution records based on museum collections (one abnormal distribution site for Pseudohynobius jinfo and Tylototriton wenxianensis, two abnormal distribution sites of Liua shihi). These abnormal sites were identified by their elevation being much lower than elevation records of relevant references and field surveys. This may be due to inaccurate description of occurrence sites of some museum collections. We excluded these abnormal sites and reanalyzed our revised dataset. Results indicate that the potential distribution areas of P. jinfo and T. wenxianensis are much smaller than those in our previous manuscript. However, this is more reasonable since these species are distributed in high elevation areas in Chongqing, which is only a small portion of the area. The updated potential distribution areas of the three salamanders are also more consistent with our knowledge of their natural history and habitat based on both field survey and relevant references. We are lucky that we have found this flaw before it might mislead the conservation on these animals, and we deeply appreciate your help.

Comments 3: Some images need to improve, either too small legends or symbols that lack contrast and are difficult to see. The title of your manuscript should be shortened and improved, and the scientific names in parentheses should be included.

Response 3: Thank you for your valuable suggestions. We have updated and redrawn the figures, and made efforts to adjust the legends and symbols accordingly. Additionally, we have included a map that illustrates the administrative divisions of Chongqing Municipality. The title has also been revised to reflect your recommendations.

Comments 4: Avoid repetitions in your text. Too often, do you mention rare and endemic.

Response 4:  Thank you for your feedback. In accordance with your suggestions, we have revised the manuscript to eliminate repetitive usage of the terms "rare" and "endemic."

Comments 5: I have no idea what Class I and II means and the more commonly known IUCN threat status should be used. For further comments and corrections see my comments in the attached PDF.

Response 5: Thank you for your insightful comments. We have made the necessary revisions to the manuscript.

Reviewer 5 Report

Comments and Suggestions for Authors

The manuscript "Potential Distribution of Three Endemic Rare Salamanders in Chongqing, China Under Climate Change and Their Conservation Implications" is well-written and provides intriguing information regarding the impacts of climate change and human activities on these species.

There are a number of geographic references that may be confusing to a reader who is unfamiliar with that region but the authors provide graphics (maps) that are helpful and the study itself is more geared toward those responsible for the management of these three species.

Specific comments and suggestions are included in the attached file.

Author Response

We greatly appreciate your dedication of time and effort in reviewing this manuscript. Thank you for your nice comments. We have revised our manuscript. Generally, we improved our title; expanded our introduction on the natural history and distribution of the three salamanders; rechecked our data and reanalyzed it, supplemented detailed description on methods; updated our results and discussion; figures were redrawn to make it more readable; languages were also improved to reduce replicates (our replies are all red-colored).

1. We have re-examined our methods and datasets manually. We have found few abnormal distribution records based on museum collections (one abnormal distribution site for Pseudohynobius jinfo and Tylototriton wenxianensis, two abnormal distribution sites of Liua shihi). These abnormal sites were identified by their elevation being much lower than elevation records of relevant references and field surveys. This may be due to inaccurate description of occurrence sites of some museum collections. We excluded these abnormal sites and reanalyzed our revised dataset. Results indicate that the potential distribution areas of jinfo and T. wenxianensis are much smaller than those in our previous manuscript. However, this is more reasonable since these species are distributed in high elevation areas in Chongqing, which is only a small portion of the area. The updated potential distribution areas of the three salamanders are also more consistent with our knowledge of their natural history and habitat based on both field survey and relevant references.

2. To address the question of how we determined the areas where the three salamanders do not distribute in Chongqing, we added data from additional extensive field work relevant to amphibians in Chongqing from July 2014 to July 2018. We have extensively surveyed entire Chongqing covering 1760 transects, and the absence of the three salamanders in the lowlands of Chongqing was confirmed by field surveys.

3. The introduction to natural histories and distribution of the three salamanders was expanded and relevant references were provided.

4. Figures were updated and redrawn. We added a map showing the administrative divisions of Chongqing Municipality.

The following are the replies to your comments:

Comments 1: The manuscript "Potential Distribution of Three Endemic Rare Salamanders in Chongqing, China Under Climate Change and Their Conservation Implications" is well-written and provides intriguing information regarding the impacts of climate change and human activities on these species.

Response 1: Thank you for your positive feedback on our manuscript. We are pleased to hear that you found the information provided to be intriguing and relevant to the conservation issues facing these species. We acknowledge the importance of understanding the impacts of climate change and human activities on endemic and rare salamanders, and we appreciate your recognition of the effort put into this research. We look forward to further discussions and are open to any suggestions that could strengthen the findings and implications of our study.

Comments 2: There are a number of geographic references that may be confusing to a reader who is unfamiliar with that region but the authors provide graphics (maps) that are helpful and the study itself is more geared toward those responsible for the management of these three species.

Response 2: Thank you for your constructive comments. We have taken your feedback into consideration and have revised the manuscript accordingly. Additionally, we have included a new map that delineates the administrative divisions of the Chongqing Municipality to assist readers unfamiliar with the region. We believe these improvements will make the study more accessible and informative, particularly for those involved in the management and conservation of the three species in question.

Comments 3: Specific comments and suggestions are included in the attached file.

Response 3: We are grateful for the specific comments and suggestions you have provided in the attached file. Each point has been carefully considered and addressed in the revised version of our manuscript. We appreciate your thorough review and believe that your insights have significantly contributed to the enhancement of our study.

Round 2

Reviewer 4 Report

Comments and Suggestions for Authors

Dear authors, I appreciate your detailed revision and response to my feedback. The introduction and methods are very clear and the new title is very good.